# A multiplexed, automated evolution pipeline enables scalable discovery and characterization of biosensors

Brent Townshend[1,4], Joy S. Xiang [1,4], Gabriel Manzanarez[1], Eric J. Hayden [1,3] & Christina D. Smolke [1,2✉]

Biosensors are key components in engineered biological systems, providing a means of measuring and acting upon the large biochemical space in living cells. However, generating small molecule sensing elements and integrating them into in vivo biosensors have been challenging. Here, using aptamer-coupled ribozyme libraries and a ribozyme regeneration method, de novo rapid in vitro evolution of RNA biosensors (DRIVER) enables multiplexed discovery of biosensors. With DRIVER and high-throughput characterization (CleaveSeq) fully automated on liquid-handling systems, we identify and validate biosensors against six small molecules, including five for which no aptamers were previously found. DRIVER-evolved biosensors are applied directly to regulate gene expression in yeast, displaying activation ratios up to 33-fold. DRIVER biosensors are also applied in detecting metabolite production from a multi-enzyme biosynthetic pathway. This work demonstrates DRIVER as a scalable pipeline for engineering de novo biosensors with wide-ranging applications in bio-manufacturing, diagnostics, therapeutics, and synthetic biology.

[1] Department of Bioengineering, Stanford University, Stanford, CA, USA. [2] Chan Zuckerberg Biohub, San Francisco, CA, USA. [3]Present address: Department of Biological Science, Boise State University, Boise, ID, USA. [4]These authors contributed equally: Brent Townshend, Joy S. Xiang. ✉email: csmolke@stanford.edu

Sensing small molecules is the foundation of many ex vivo and in vivo applications, in both natural and synthetic systems. The ability to detect and measure diverse chemicals and biomarkers is instrumental in disease diagnosis, prognosis, and treatment, as well as in monitoring the current state of a healthy system or the environment. Furthermore, sensing is a prerequisite to control. Sensors coupled to biological actuators, such as transcription or translation control devices, are the basis of many natural biological control systems[1]. Similarly, engineered genetic devices that sense drugs or metabolites can precisely and dynamically control gene expression inside cells[2]. Ultimately, achieving programmable biocomputation and large-scale biological circuit design will rely on the availability of sensor–actuator elements to transduce diverse signals from a large biochemical space for complex phenotypic control[3–7].

A desirable sensing platform is one in which concentrations of a broad range of analytes can be detected and measured using a common framework. Although sensing based on specific physical or chemical properties of an analyte has been widely deployed, the range of targets is limited and the detection methods unique to each. Transcription factor-based biosensors are commonly used in ligand-inducible gene regulation control[4], but lack a systematic framework for developing new sensing capabilities. Antibody-based biosensors[8] provide such a framework, but are not effective at sensing small molecules and are not amenable to in vivo gene expression control. Nucleic acid aptamer-based biosensors are emerging as an attractive alternative that can sense a wide range of analytes, bind with high affinity and selectivity, are readily manufactured in vitro or in vivo, and do not require costly cold chain storage. Furthermore, systematic methods exist for aptamer discovery such as SELEX (systematic evolution of ligands by exponential enrichment), in which large nucleic acid libraries are iteratively enriched for high-affinity aptamers de novo to bind proteins or small molecules[9–11]. In addition, RNA aptamer-based biosensors have been demonstrated in a variety of biological systems for conditional gene expression control[12–17].

While aptamers have been used extensively for protein biomarker measurement[18], their application to sense small molecules and other chemicals or as sensors coupled to biological actuators in vivo has been more limited[19]. This is primarily due to the lack of high-throughput selection methods that generate sensors that can function in vivo. Existing methods for developing aptamer-based biosensors fall into two broad categories: those that find aptamers alone for the target based on binding and those that aim to discover aptamers in the context of a biosensor. In the former category, conventional SELEX approaches[10] require separation of the bound aptamer–ligand complex from unbound library members, which usually entails chemical modification of the ligand (Fig. 1a). Such modifications are problematic as they are molecule-specific, preclude selection against complex mixtures, and may result in the generation of aptamers that do not bind the unmodified molecule (Fig. 1e). One recently described approach, Capture-SELEX[20,21], is unique in that it does not require chemical modification of the ligand; however, the method depends upon specific binding-mediated conformational changes to separate binders, a requirement separate from any conformational changes needed to ensure in vivo functionality of a biosensor. Furthermore, all of the methods that select for an isolated aptamer do not directly form the basis of an in vivo biosensor without further development and optimization[22]. In the second category are methods that jointly select for aptamers in the context of a biosensor platform, all of which are based on integration of the aptamer with a ribozyme where binding modulates cleavage[23,24]. Separation then depends upon polyacrylamide gel electrophoresis (PAGE) to isolate only cleaved or uncleaved fractions of the population. The incorporation of a gel-based separation approach

into a method based on iterative cycles of selection introduces a manual and time-consuming step. In addition, during selection rounds performed with ligand present, the library will renature in the absence of ligand following gel separation. These methods depend upon chelation of $Mg^{2+}$ to prevent cleavage during this step, but biosensors that operate in low-$Mg^{2+}$ concentrations will nonetheless cleave and be removed from the selection. Thus, gel-separation-based methods have limited utility in achieving biosensors intended to operate in the biologically relevant low-$Mg^{2+}$ levels in vivo. As a result, to our knowledge, no biosensors developed using PAGE-separation-based joint-discovery methods have been shown to directly function in vivo.

We developed a scalable method to address limitations in biosensor development: DRIVER (De novo Rapid In Vitro Evolution of RNA biosensors) provides a fully automatable selection method in the context of an RNA biosensor platform, allowing the entire process to be carried out in solution, without modification of ligands (Fig. 1d). A critical library regeneration step was introduced to significantly improve the efficiency of the method such that selections could be performed fully in solution and automated on liquid handling robots. Using our platform, we generated multiple biosensors with nanomolar to micromolar sensitivities to each of six diverse small molecules. We also developed CleaveSeq, a high-throughput assay based on next-generation sequencing (NGS), that was used to characterize newly evolved biosensors in parallel by counting cleaved and uncleaved reads for each sequence in a mixed library (Fig. 1f). Finally, selected biosensors were directly applied to regulate gene expression in live cells, indicating the potential of the platform to generate biosensors for wide-ranging in vitro and in vivo biosensing applications.

## Results

**A regeneration method that enables solution-based separation of RNA biosensors.** DRIVER and CleaveSeq employ a regeneration method central to the effectiveness of the selection process and accuracy of the functional assay. In both methodologies, biosensor libraries are transcribed and undergo cleavage with or without ligands. To use the cleaved product for a subsequent round of selection, the 3′-cleaved ribozyme products that contain the library diversity, including the nucleotides responsible for biosensor function, must have their 5′ end regenerated. Efficient regeneration is essential to avoid biases in the selection process unrelated to target binding and to maintain diversity. Although prior methods[25,26] have employed separate splint oligonucleotides to improve ligation, substantial biases likely exist, especially where secondary structure is present at the ends to be ligated, and, if used for selection, lead to selection bias, generation of undesirable amplicons, and reduction of diversity. We achieved unbiased regeneration through a unique triple-function oligonucleotide that combines a reverse transcription (RT) primer with a ligation substrate and a splint sequence resulting in much higher local concentrations of the ligation components (Fig. 2b and Supplementary Figure 1). Following a cleavage step where a DNA library is transcribed to RNA and constituent ribozymes allowed to self-cleave, this oligonucleotide is used in a RT step to create a complementary DNA (cDNA) product. The same oligonucleotide then guides the repair of the cleavage-induced changes during a ligation step, resulting in a ligation product with a different 5′ prefix from cDNA that arose from uncleaved RNA. Finally, PCR is used to selectively amplify the cDNA corresponding to either the cleaved or uncleaved library members (Fig. 2b). In DRIVER, this product is then used for the next round of a selection. For high-throughput measurement of cleavage of a library of biosensors, CleaveSeq uses the distinct regenerated

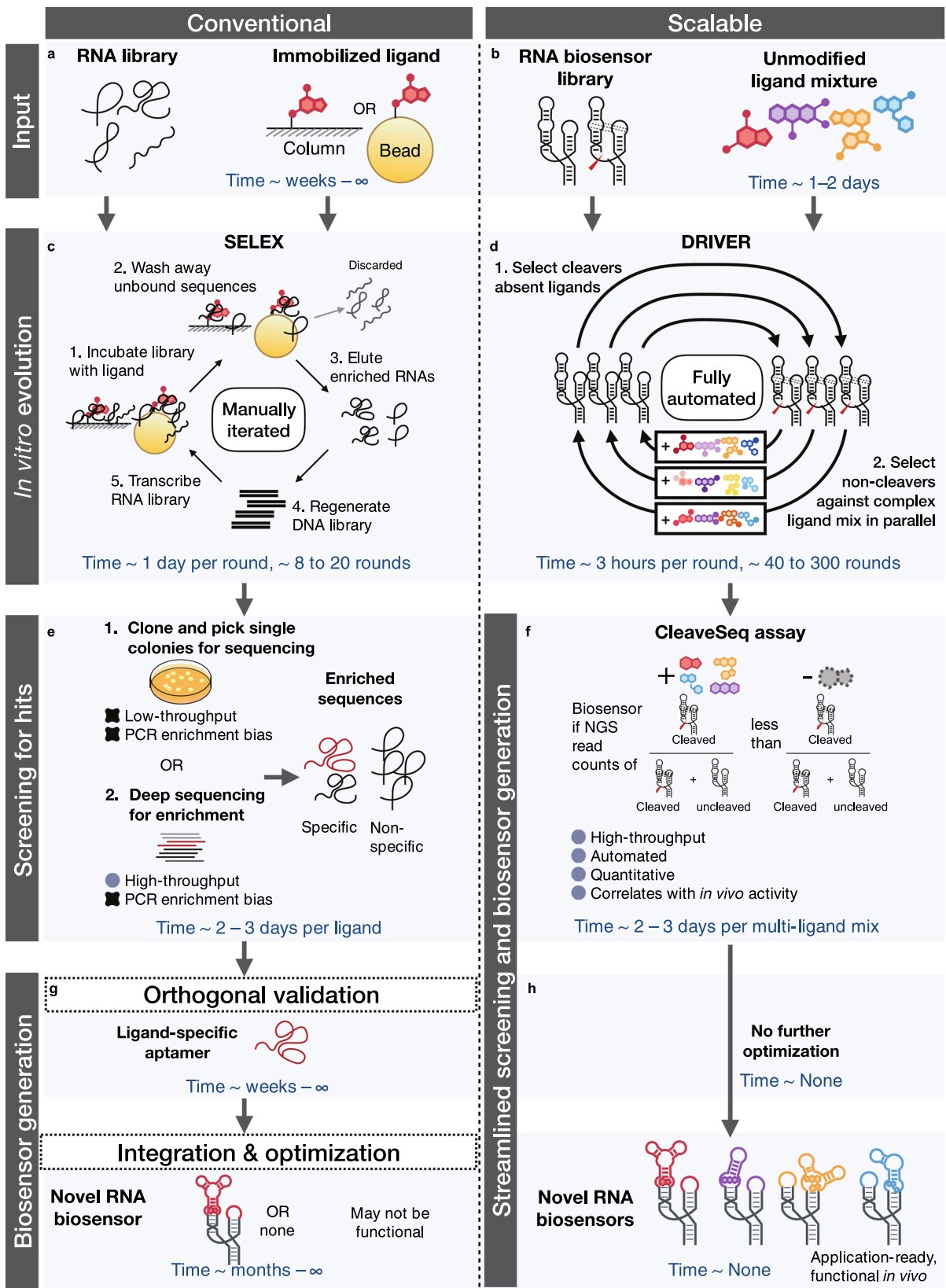

prefix to distinguish between cleaved and uncleaved in NGS read counts (Fig. 2c).

**DRIVER enables solution-based de novo selection of biosensors.** DRIVER places the regeneration procedure within a

directed-evolution context (Fig. 3a) to select for sequences that exhibit higher cleavage in the absence of ligand and lower cleavage in the presence of ligand. Rather than using physical separation to enrich binding members of a library as is typically done in SELEX and PAGE-based ribozyme selections[23,24], DRIVER leverages the sequence modifications resulting from the

**Fig. 1 De novo Rapid In Vitro Evolution of RNA biosensors (DRIVER) is a scalable platform that allows automated, parallelized selection of novel biosensors to diverse ligands. a** Conventional selection starts with an RNA library with randomized regions in an unstructured context and chemical conjugation of a single ligand to a solid-phase support such as columns or beads. **b** Scalable DRIVER selection begins with an RNA biosensor library with randomized loop regions in a self-cleaving ribozyme framework and conjugation-free ligands as complex mixtures in solution. **c** Conventional SELEX (systematic evolution of ligands by exponential enrichment) involves manually iterating through 1. library incubation with a single immobilized ligand, 2. washing away unbound sequences, 3. eluting enriched sequences, and 4. regenerating the DNA library template and transcribing the RNA library. **d** DRIVER selection is fully automated with liquid handling, alternately enriching for sequences that co-transcriptionally self-cleave in the absence of ligands, and remain uncleaved in the presence of ligands. Liquid handling enables selection against complex mixtures in parallel, continuously generating hits across different rounds. **e** An enriched library from conventional SELEX can be transformed into bacteria to isolate sequences from single colonies, or deeply sequenced with next-generation sequencing (NGS) to identify enriched sequences. **f** DRIVER-evolved libraries are screened using the fully automated CleaveSeq assay, which uses NGS to identify sequences that are cleaved and uncleaved in the absence and presence of ligand, respectively. **g** Hits from conventional SELEX require further orthogonal assays to verify ligand-specific binding versus PCR amplicons or non-specific binders to the column or bead. Once verified, extensive optimization is usually required to convert ligand-specific aptamers into biosensors, with no guarantee of success. **h** DRIVER hits are directly functional in live cells, requiring no further optimization. Time on individual panels indicates an estimated duration for each set of experimental procedures, and ∞ indicates that the desired outcome is not achieved.

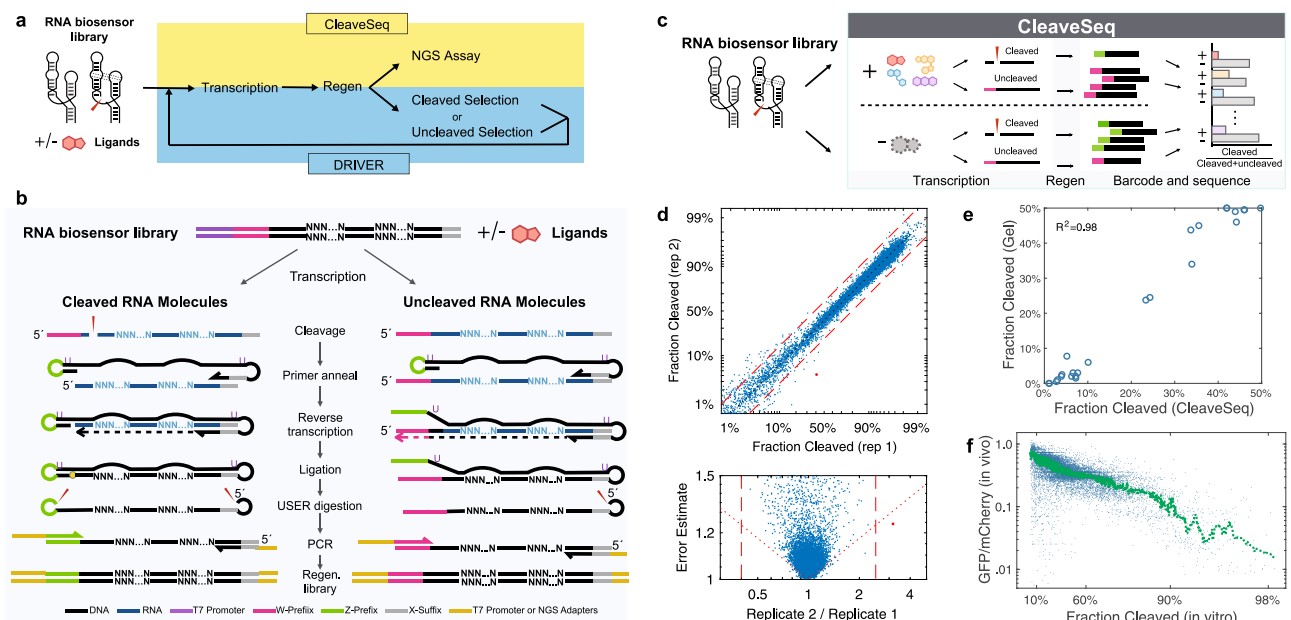

**Fig. 2 Regeneration of ribozymes after cleavage enables selection and an NGS-based assay that are correlated with in vivo activity. a** CleaveSeq and DRIVER use the same method to regenerate the 5′ prefix after cleavage. **b** The regeneration method selectively restores the 5′ cleaved portion of the ribozyme and replaces the prefix sequence with a new prefix (e.g., "W" prefix is replaced with "Z" prefix for cleaved RNA molecules). The process starts with co-transcriptional cleavage of a DNA template library. An oligonucleotide is then added and annealed to the 3′ end of the resulting RNA pool for reverse transcription. The oligonucleotide subsequently hybridizes to the nascent cDNA, forming a partially self-annealing double-stranded hairpin that brings together the ends of molecules derived from the cleaved RNA, enhancing self-ligation. The circularized ligation product is cut at two uracil locations by Uracil-Specific Excision Reagent (USER), releasing a linear DNA strand harboring the desired sequence with a new prefix sequence. Two distinct populations of DNA molecules result: those corresponding to RNA that did not cleave and those that correspond to cleaved RNA, the latter of which will have the new prefix. One population is selectively PCR-amplified with primers that extend the product with either the T7 promoter (for DRIVER) or NGS adapters (for CleaveSeq). **c** CleaveSeq measures the relative abundance of cleaved and uncleaved molecules to provide estimates of cleavage fractions and switching for each library sequence. **d** Representative comparison of cleavage fractions for two replicates independently carried through the CleaveSeq assay ($N = 12{,}025$, at least 100 reads/sequence in each analysis). Bottom panel shows the standard error of the ratio on the y-axis. Significant (two-sided test with Bonferroni correction: $p < 0.1/N$) outliers are shown as red dots. Red dashed lines delineate 2.5-fold change of cleavage. **e** Comparison of CleaveSeq assay and gel electrophoresis analysis. Twenty-one biosensors, the hammerhead ribozyme, and a non-cleaving control were analyzed via CleaveSeq and on a 10% PAGE gel (Supplementary Figure 5). **f** Comparison of in vitro cleavage fraction (CleaveSeq) and in vivo gene regulatory activity. Each point represents an individual biosensor sequence. Gene regulatory activity, measured as the ability of the biosensor to control *GFP* reporter expression in yeast in a previously reported FACS-Seq assay[27] ($N = 16{,}699$). Green line is data processed by a 835-point median filter. Source data is available in the Source Data file.

ligand-dependent cleavage activity to enrich for ligand-binding sequences (or nonbinding ones for negative selections) using PCR. Thus, DRIVER avoids the need to modify the target molecule for binding to an affinity matrix and allows all steps of selection to be performed in solution against any soluble target ligand or even complex mixtures; the latter permitting parallel selections. Ribozyme-based selections enrich slowly due to the existence of non-switching sequences that can fold into cleaving or non-cleaving conformations and thus have up to 50% survival in each selection round[23]. However, DRIVER makes many rounds of selection feasible as all of the steps in DRIVER require only liquid movements and thermocycling and the entire

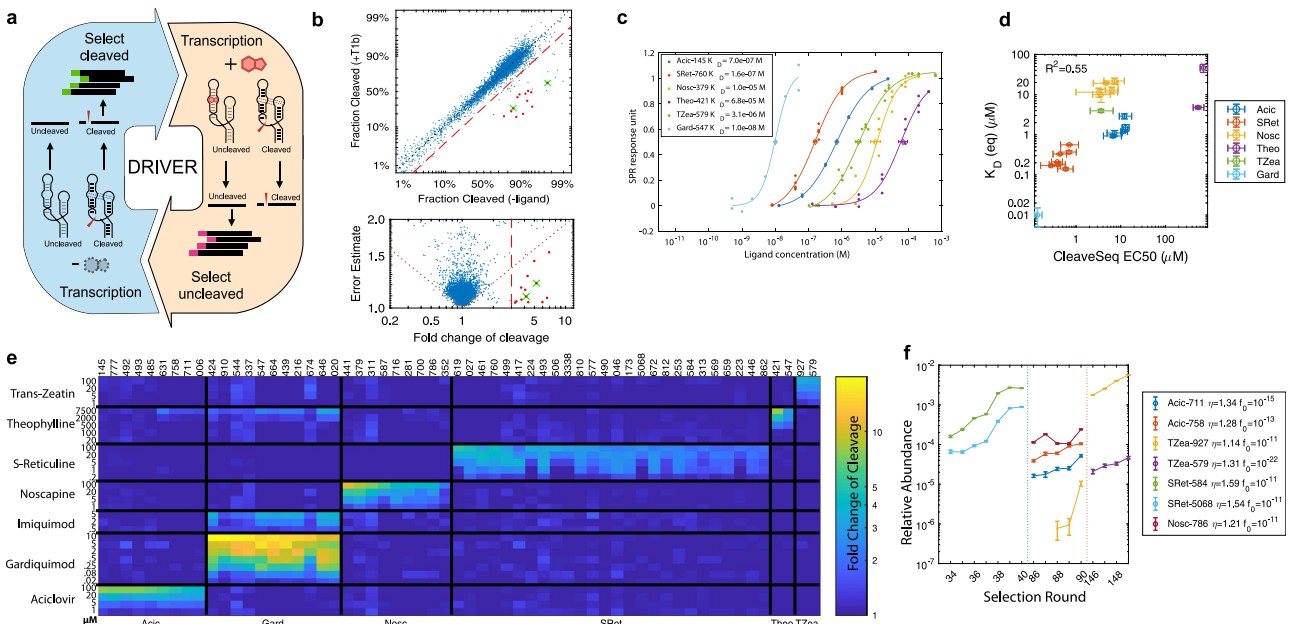

**Fig. 3 Biosensors from DRIVER exhibit high affinities and selectivities. a** A high-diversity DNA library of potential RNA biosensors are used as inputs to DRIVER. For each round of selection, the library is transcribed to RNA. For rounds with ligands present (right yellow shaded workflow arrow) the cDNA corresponding to uncleaved product is amplified with a PCR primer specific to the prefix attached to the input to the current round. For selection rounds without ligand present (left blue shaded workflow arrow), the cleaved product is prepended with a new prefix using the regeneration method shown in Fig. 2 and is amplified using a PCR primer specific to this prefix. **b** Cleavage fractions in the presence and absence of the T1b ligand mixture for a DRIVER-enriched library at round 74 ($N = 4328$; at least 30 reads/sequence in each condition). Bottom panel shows the standard error of the fold change in cleavage fractions. Dotted lines delineate the region where a multiple-hypothesis test ($\alpha = 1/N$) would reject the null hypothesis of non-switching; dashed line shows where the fold change of cleavage is 3×; red dots indicate sequences with strong (>3×), significant switching; green crosses indicate validated biosensors that were first identified from this analysis. **c** SPR binding dose–response curves of select biosensors. Diamonds mark the equilibrium dissociation constants $K_D$, as mean and s.e.m. of $n = 3$ biologically independent experiments (except for Gard-547, as noted in "Methods"). **d** Comparison of equilibrium $K_D$ and CleaveSeq $EC_{50}$ for select biosensors. $K_D$ are shown as mean and error bars in s.e.m of at least $n = 3$ biologically independent experiments. $EC_{50}$s are based on triplicate assays over the ligand and concentrations shown in (**e**). **e** Fold change of cleavage fractions for select biosensors measured via CleaveSeq (Supplementary Data 3). Columns represent different biosensor sequences, grouped by ligand. Rows represent ligand concentrations used in the assay. The color represents the fold change of cleavage between the – and + ligand conditions. See also Supplementary Figure 6. **f** NGS determined the relative abundance of select biosensors during selection rounds immediately prior to their discovery. Data are presented as estimates of the binomial proportion ± standard deviation. The legend indicates the average enrichment/round and extrapolated round 0 fractions based on exponential fits to these data. Source data are provided as a Source Data file.

selection process has been automated on a liquid-handling robot that can run continuously at 8–12 rounds/day, allowing fully parallelized selections to be run in 1–2 weeks.

We started selections with a high-diversity library ($10^{12}$–$10^{14}$) of potential sensors formed by expanding and randomizing the loop sequences of a hammerhead ribozyme (Supplementary Figure 2). This design is based on the expectation that ligand binding to specific sequences at the larger randomized loop, where aptamers are likely to arise, modulates the extent of ribozyme self-cleavage by interfering with interactions between that loop and the smaller randomized loop[27]. Each round of evolution selectively enriches a subset of sequences from the library. During rounds with ligand(s) present, the RNA molecules with an unmodified prefix sequence from the prior round, corresponding to sequences that did not cleave, are PCR amplified using the prefix and a fixed suffix sequence as priming sites (Supplementary Table 2). For negative selection rounds (without the target ligand(s) added), the molecules with the ligated prefixes, corresponding to RNA molecules that underwent self-cleavage, are amplified using the newly added prefix and fixed suffix as primers. By iteratively applying this process, DRIVER enriches self-cleaving RNA sequences that are sensitive to the ligand(s) in the mixture. At intervals of 32–40 selection rounds, the products were constricted by dilution to 10,000–50,000

unique sequences and analyzed via CleaveSeq to identify any enriched sequences responsive to target ligands (Fig. 3b and Supplementary Figure 3).

Four selection runs (S1–S4) were performed against a panel of small molecules, including plant hormones, secondary metabolites, and small-molecule therapeutics. As part of the method development, each run had variations in certain parameters such as ligation method, magnesium concentration, and the ratio of positive to negative selection rounds (Supplementary Table 1). S1 was a preliminary selection run used to develop and optimize the enrichment methods, using theophylline as the target ligand. The selection was run for 57 rounds, and the CleaveSeq assay was applied to the resulting enriched libraries to identify potential sensors. The selected sequences were synthesized from oligonucleotides and assayed individually in the presence and absence of the ligand mixtures to determine the mapping between specific sequences and their ligands. We identified nine distinct sensors for theophylline, all with sequences distinct from previously published aptamers (Supplementary Figures 3a and 4a). S2 was run using three different ligand mixtures that underwent independent selections in separate wells, with the same selection operations applied in parallel. The ligand mixtures were chosen based on the compatibility of buffers for the suspension of 13 different small molecules (Supplementary Data 1). The

S2 selection was run for 98 rounds and nine distinct sequences sensitive to (S)-reticuline, a molecule for which no prior known aptamer existed, were identified (Supplementary Figures 3b and 4b). Based on the results of S1 and S2, several parameters were adjusted to improve enrichment (Supplementary Table 1). S3 was run against three ligand mixtures composed of 45 different small molecules. By the end of selection S3, the three selection mixtures had seen 114, 202, and 198 rounds of selection, with multiple sensors identified to (S)-reticuline, noscapine, *trans*-zeatin, and aciclovir (Fig. 3b and Supplementary Figure 3c–e). Finally, S4 was run against a subset of the S3 ligand mixtures using a 1:2 ratio of positive and negative selection rounds to bias selection towards producing sensors with higher cleavage fractions. After 102 rounds, several sensors sensitive to gardiquimod were isolated. Of the six ligands for which sensors were found in the four selection runs, only theophylline had a previously identified aptamer.

We retrospectively determined enrichment rates of validated RNA biosensors and compared these with the theoretical rates and number of rounds required to enrich the sensors to detectable levels. By sequencing the DNA library present at the end of each of the four or more selection rounds prior to the round where each biosensor was first observed, we were able to determine the fractional concentration of the biosensor as a function of round number and thus its enrichment rate (Fig. 3f). We found enrichment rates ranging from 1.2 to 1.6 × /round, with the most highly enriching sequences showing up in earlier rounds of the selection. By projecting back from the fraction, $f_n$, of a sequence present at round $n$ using the measured enrichment rate, $\eta$, and assuming that the bulk cleavage fraction of the library remains relatively constant and that the sequence was present in the starting library, we estimate the fraction of the biosensor in the initial library, $f_0$ (Fig. 3f). The values for $f_0$ generally range from $10^{-11}$ to $10^{-9}$, consistent with the starting diversity of the library. A few sensors exhibit significantly lower $f_0$ values; these likely arose from a mutation during an intermediate selection round rather than having been present in the initial library, indicating that the accessible diversity is likely higher than the starting library size due to mutations throughout the process. Taken together, these results are consistent with our models for enrichment, mutations, and initial library diversity.

**High-throughput characterization of biosensor libraries indicate high ligand sensitivity and selectivity.** In CleaveSeq, both the cleaved and uncleaved library members are amplified and adapters are added in the PCR step such that NGS can be used to quantitatively measure the relative abundance of these populations, and thereby the fraction of RNA molecules that underwent cleavage, for each distinct sequence in the template library (Fig. 2c). The impact of ligands on the cleavage fraction of each sequence can be measured by performing the assay under different conditions with a single ligand or with a ligand mixture. CleaveSeq can thus be used to identify sequences within a library that are sensitive to particular ligands or mixtures, for example, after a set of selection rounds.

To demonstrate the reproducibility of the CleaveSeq assay, we performed replicate measurements on a panel of ribozyme and ribozyme-based biosensors with known activities that span a range of cleavage fractions (Fig. 2d and Supplementary Data 3). The replicates agreed (the null hypothesis that the replicas were identical could not be rejected at the $p = 0.1/\text{N}$ ($8 \times 10^{-6}$) level, where $N$ is the number of sequences measured) for all but one sequence. Further analysis showed that the single outlier sequence was one nucleotide different from other sequences that occurred at least 100× more frequently—a situation likely due to PCR

mutations during the NGS library preparation that can result in assigning read counts to an incorrect sequence.

We further validated CleaveSeq by comparing the cleavage fractions obtained through the NGS-based assay with traditional gel-based cleavage assays. Twenty-three RNA sequences harboring ribozymes, including the wild-type hammerhead ribozyme and non-cleaving controls, were incubated in a cleavage reaction and PAGE was used to analyze the resulting RNA. The intensity of bands corresponding to the uncleaved and the 3′-cleavage products was compared to estimate the fraction of molecules that underwent cleavage for each sequence (Supplementary Figure 5). A mixture of this same set of sequences was assayed using CleaveSeq to produce a cleavage fraction estimate for each sequence under similar transcription conditions. The data demonstrate high correlation between the two assays (Fig. 2e).

To more extensively characterize sensors identified from DRIVER selections, their DNA templates were individually synthesized, mixed at equimolar concentrations, and evaluated with CleaveSeq under various ligand and ligand mixture concentrations. These biosensors were mixed with 42 non-switching controls, including the native hammerhead ribozyme (satellite RNA of the tobacco ringspot virus (sTRSV)), an inactive mutated ribozyme (sTRSVctrl), and ribozyme variants that exhibit a range of ligand-independent cleavage fractions[27]. The CleaveSeq products were barcoded at the PCR stage and sequenced to obtain counts of cleaved and uncleaved RNA molecules for each condition and sequence. Analysis of the read counts provided measurements of cleavage fractions for each condition and thus the sensitivity as measured by the fold change of cleavage fraction for each of the ligand and sequence conditions (Fig. 3e and Supplementary Data 3). These data were also used to derive half-maximal effective concentration ($EC_{50}$) values for each sensor (Supplementary Figure 6). The most sensitive sensors to aciclovir had an $EC_{50}$ of 5 µM, gardiquimod 80 nM, noscapine 5 µM, (S)-reticuline 1 µM, theophylline 500 µM, and *trans*-zeatin 5 µM, indicating that DRIVER can select for biosensors with high affinities to ligands.

The selectivity of each biosensor was assessed using the CleaveSeq assay in the presence of each of several molecules structurally related to the expected ligand at identical concentrations. This analysis provided an estimate of the fold change of cleavage relative to the no-ligand condition for each sensor–molecule combination (Supplementary Figure 7). In general, the biosensors were found to be more sensitive to their expected ligand than other tested ligands, with the exception of the aciclovir biosensors, which exhibited higher sensitivity to ganciclovir (4.0–4.8-fold) than to aciclovir (2.8–4.0-fold) (Supplementary Figure 7a); ganciclovir contains an additional alcohol group in its side chain that acyclovir lacks. However, the aciclovir biosensors were less sensitive to valacyclovir (1.4–2.8-fold) and insensitive to famciclovir, which both have larger side chains than either acyclovir or ganciclovir. The theophylline biosensor was tested against the closely related methylxanthines caffeine and theobromine (Supplementary Figure 7b). While theophylline and theobromine are structural isomers that both possess two methyl groups, theophylline on the 1 and 3 position nitrogens and theobromine on the 3 and 7 position nitrogens, caffeine possesses three methyl groups on the 1, 3, and 7 position nitrogens. Caffeine elicited no response and theobromine elicited a lower response (2.2-fold) than theophylline (2.8-fold). All of the gardiquimod sensors showed at least six times greater sensitivity to gardiquimod than to the related compounds resiquimod and imiquimod (Supplementary Figure 7c). Gardiquimod contains a 1*H*-imidazo[4,5-*c*]quinoline-4-amine core with two side chains, one containing a secondary amine and the other a hydroxylated isobutyl group. Resiquimod is identical to gardiquimod except for the substitution of oxygen for the nitrogen in the secondary amine side chain, forming an ether linkage in its

place, while imiquimod lacks the secondary amine side chain entirely, as well as the hydroxyl group on the isobutyl side chain. The *trans*-zeatin sensors showed 1.5 times higher sensitivity to *trans*-zeatin, an adenine derivative with a hydroxylated isoamylene side chain on the adenine 6-amine, than to the related compounds 6-benzylaminopurine and kinetin, which have benzyl and furfuryl side chains, respectively, in place of the hydroxylated isoamylene group (Supplementary Figure 7d). For the two benzylisoquinoline alkaloid (BIA) targets, noscapine and (*S*)-reticuline, we chose seven commercially available BIAs and precursors to compare against (Supplementary Figure 7e, f). The noscapine sensors did not exhibit sensitivity to (*S*)-reticuline or the other BIAs tested; however, none of the tested BIAs share the same phthalideisoquinoline backbone as noscapine. In contrast, two of the BIAs tested, norlaudanosoline and norcoclaurine, share the same 1-benzylisoquinoline backbone as (*S*)-reticuline. The (*S*)-reticuline biosensors exhibited some sensitivity to norlaudanosoline (1.0–3.0-fold), and none to norcoclaurine, but all exhibited higher sensitivity to (*S*)-reticuline (3.5–5.5-fold). While both norlaudanosoline and norcoclaurine lack the three methyl groups that decorate the 1-benzylisoquinoline scaffold in (*S*)-reticuline, norlaudanosoline and (*S*)-reticuline both have a hydroxyl group present on the benzyl moiety that norcoclaurine lacks, indicating that this functional group may play a role in sensor binding. The results of this analysis also indicate that all the biosensors have little or no cross-selectivity to the other ligands used in the DRIVER selections at comparable concentrations (Fig. 3e). Although the tested compounds necessarily represent a small sampling of the possible molecular space to which these biosensors may respond, these results taken together indicate that DRIVER-selected biosensors recognize specific regions of the small-molecule ligand, which can be shared among similar chemical compounds, but that relatively small differences in chemical structure can be distinguished.

For a subset of the DRIVER-selected sensors, we further validated the binding affinities using a previously described surface plasmon resonance (SPR) assay at physiological $Mg^{2+}$ concentration of 0.5 mM[28] (Fig. 3c, Supplementary Figure 8, and Supplementary Data 4). The gardiquimod and (*S*)-reticuline biosensors exhibit high-binding affinities with equilibrium dissociation constants ($K_D$) in the nanomolar range, with Gard-547 exhibiting the tightest binding at a $K_D$ of 10.1 nM and SRet-760 at a $K_D$ of ~160 nM. The $K_D$ of aciclovir sensors and the *trans*-zeatin sensor TZea-579 are generally in the high nanomolar to low micromolar range (680 nM to 3.6 μM), whereas the $K_D$ of the *trans*-zeatin sensor TZea-927, noscapine, and theophylline sensors are approximately an order of magnitude higher (10–72 μM). Kinetic on and off rates ($k_{on}$ and $k_{off}$) were also measured for reticuline, aciclovir, and *trans*-zeatin biosensors (Supplementary Figure 8a–f). Dissociation constants, $K_D$, derived from $k_{on}$ and $k_{off}$ for these sensors are also found to be similar to the equilibrium $K_D$ measurements (Supplementary Figure 8g and Supplementary Data 4). In comparison to the biophysical measurements, CleaveSeq determined the $EC_{50}$ of ligand-induced fold change of fraction cleaved to range from 135 nM to 700 μM (Fig. 3e), which, even though it measures cleavage activity rather than just binding, tracks the SPR-determined dissociation constants $K_D$ (Fig. 3d).

Further analysis of the binding characteristics and binding domains highlight the potential of aptamers derived from DRIVER biosensor for integration into other RNA devices for in vivo applications. We examined SPR binding affinity of minimized RNA biosensor sequences, which showed that the binding domains of some DRIVER-selected sensors are isolatable to the randomized region in the large stem loop (Supplementary Figure 9a). Specifically, the binding affinities of the Acic-145, Nosc-441, Theo-421, and TZea-579 sensors,

which are representative of their respective ligand classes, were not affected when the small stem-loop sequences were randomized. Further truncation of Acic-145, Nosc-441, and TZea-579 to just the large stem loop preserved the binding affinities, whereas truncation of Theo-421 indicated that the aptamer domain for that biosensor spans most of stem-loop II and the catalytic core (Supplementary Figure 9b). These smaller aptamer domains may be good candidates for incorporation into alternative RNA control architectures, such as CRISPR-guided RNAs[29], splicing *cis*-regulators[30], and microRNAs[31]. In contrast, the binding domains of TZea-927 and SRet-760 appear to be encoded within the randomized regions in both stem loops, requiring sequences in both ribozyme loops, likely to coordinate tertiary interactions for ligand binding (Supplementary Figure 9). It was recently highlighted that complex tertiary structures are a recurring motif of natural riboswitches for stabilizing ligand interactions and for greater robustness in activity in cellular environments[32].

In addition to strong equilibrium binding, (*S*)-reticuline biosensors exhibit slower off rates ($k_{off}$ ~5–16 m s$^{-1}$) compared to other biosensors. This may explain in part the greater activation ratios of the (*S*)-reticuline switches in vivo. Slow off rates have previously been observed to be characteristic of binding domains in naturally occurring riboswitches, whereas artificially selected aptamers tended to have faster off rates[28]. Taken together, the data suggest that the DRIVER selection platform has the capacity for generating RNA biosensors that function in physiological environments.

**DRIVER-selected biosensors regulate gene expression in cells in a ligand-dependent manner**. We demonstrated the ability of biosensors selected under in vitro conditions of DRIVER to exhibit activity for regulating gene expression in vivo as RNA switches (Fig. 4a). We used a previously reported two-color assay[27], where switches are inserted into the 3′-untranslated region (UTR) of one reporter gene (*GFP*), while the second reporter gene (*mCherry*), present on the same plasmid, is used to normalize cell-to-cell variations in fluorescence. Under these conditions, ribozymes or ribozyme-based biosensors with higher cleavage fractions will result in increased mRNA degradation and exhibit lower GFP fluorescence. If ligand binding to the RNA results in lower cleavage fractions, the transcript and reporter expression levels will be increased.

To first demonstrate that in vitro cleavage fractions are correlated and relevant to their in vivo gene-regulatory activities as RNA switches, we examined the mapping between CleaveSeq-measured cleavage fractions and gene expression in vivo in yeast. We first used a large set of sequences that had been previously characterized with fluorescence-activated cell sorting (FACS-seq)[27] and characterized the same set of sequences using CleaveSeq (Fig. 2f). The data show that in vivo expression in yeast is predominantly inversely correlated with the fraction of cleaved molecules and indicate that CleaveSeq results can be used to estimate the expected in vivo switching characteristics of an RNA biosensor in response to intracellular concentrations of the cognate ligand. We then verified this for the DRIVER-selected biosensors by comparing the cleavage activities of the biosensors in CleaveSeq with their in vivo gene-regulatory activities using flow cytometry assays. CleaveSeq-determined cleavage fractions show a similar reciprocal relationship with flow cytometry-determined in vivo gene expression regulation activity of DRIVER-selected biosensors (Fig. 4k). Since DRIVER uses the same conditions and core regeneration method as CleaveSeq, DRIVER-selected biosensors are evolved in a fitness landscape that is thus correlated with in vivo activity, resulting in directly functional in vivo biosensors.

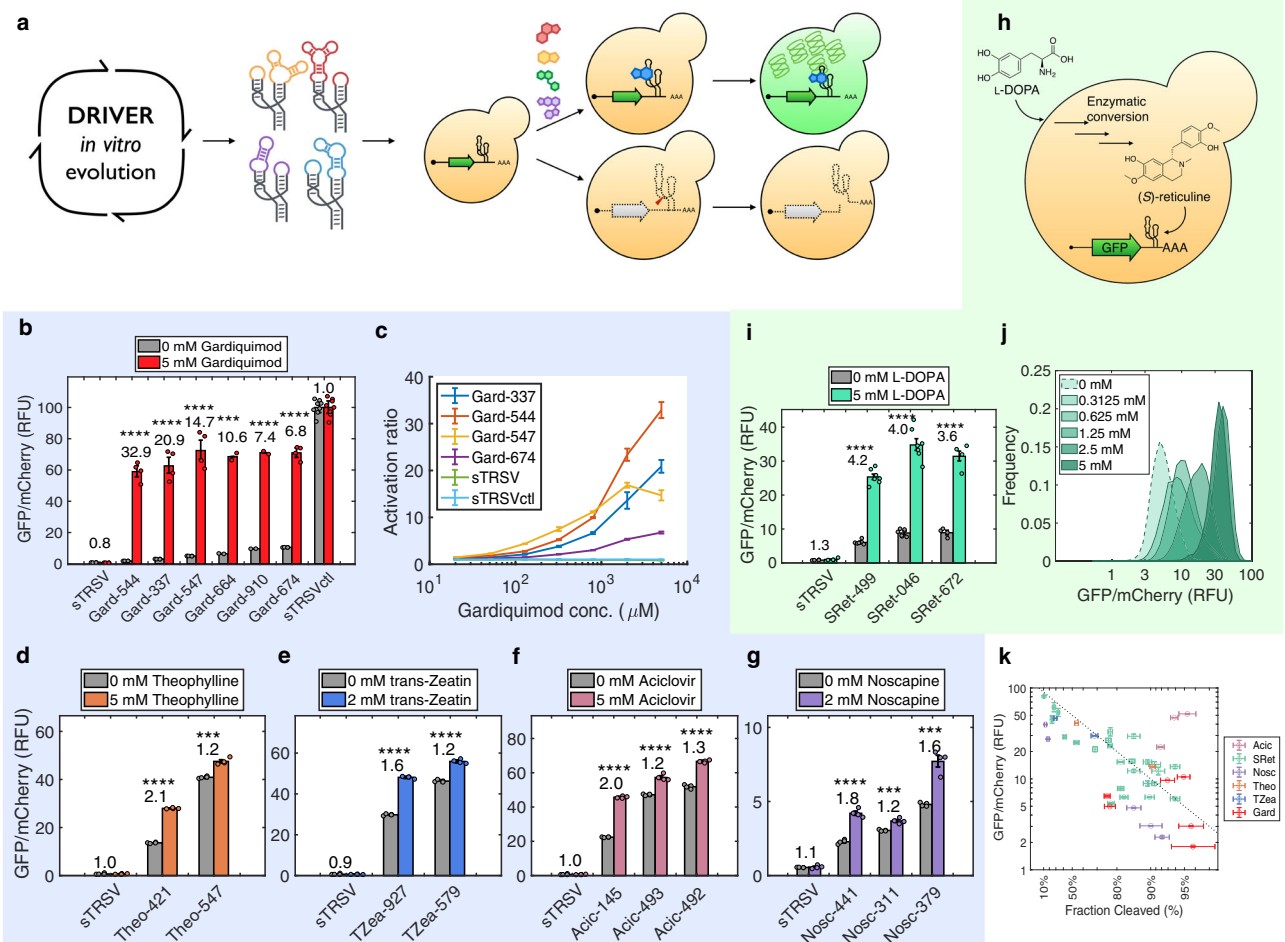

**Fig. 4 DRIVER-selected biosensors function directly as in vivo gene-regulatory switches. a** Biosensors evolved from DRIVER are validated to regulate reporter gene expression in yeast cells in response to exogenous ligand addition. Flow cytometry measurement of a fluorescent reporter under the control of **b** gardiquimod, **d** theophylline, **e** *trans*-zeatin, **f** aciclovir, and **g** noscapine biosensors in yeast in the presence and absence of exogenously added ligands. Each biosensor and control was tested in $n = 3–8$ biologically independent samples, with error bars corresponding to the standard error of mean across the replicates. Mean GFP/mCherry was normalized such that an inactive ribozyme control (sTRSVCtl) was set to 100 RFU (relative fluorescence unit). Biosensors exhibiting significant changes in gene-regulatory activity in response to their cognate ligand are denoted with asterisk(s), where ***p value <1e −5, ****p value <1e −6, from two-tailed, unpaired t tests between the with and without ligand conditions. Error bars are the standard error of the mean over $n = 3–8$ biologically independent samples for **b–g**, **i**, **k**, where individual dots indicate measurements for individual replicates. **c** Dose–response curves of activation ratios at varying concentrations of ligand for gardiquimod switches as measured by flow cytometry. **h** Biosensors evolved from DRIVER are validated to regulate reporter gene expression in response to endogenous production of a metabolite through a heterologous pathway. **i** Flow cytometry measurement of a fluorescent reporter under the control of (S)-reticuline biosensors in yeast engineered with a heterologous metabolic pathway for synthesizing (S)-reticuline from L-DOPA. L-DOPA is fed to the yeast cells to increase the production of (S)-reticuline. **j** Density plot showing the distribution of relative fluorescence levels across a population of yeast cells harboring a fluorescent reporter under the control of SRet-499 as in (**c**) under different concentrations of L-DOPA. **k** Comparison of in vitro cleavage fraction determined via CleaveSeq and in vivo gene regulatory activity via flow cytometry in yeast in the absence of ligand. Each point represents an individual biosensor sequence. Error bars are the standard error of the mean of measurements over at least $n = 3$ biologically independent experiments in each dimension. The dotted line shows the expected relationship assuming that *GFP* expression is proportional to the fraction of the mRNA that does not undergo cleavage. Source data are provided as a Source Data file.

To apply the DRIVER-selected RNA biosensors as gene-regulatory switches, biosensors to gardiquimod, theophylline, aciclovir, *trans*-zeatin, and noscapine were cloned into the 3′-UTR of the GFP fluorescent reporter. The constructs were transformed into yeast cells and the gene-regulatory activities were assayed via flow cytometry and reported as relative reporter levels (GFP/mCherry) in the absence and presence of ligand added to the growth media. The six tested gardiquimod switches exhibited large activation ratios (ratio of GFP/mCherry in the presence and absence of ligand), ranging from 6.8- to 33-fold (Gard-544) in response to 5 mM gardiquimod (Fig. 4b). These switches also exhibit graded response in activation ratios across a range of

concentrations of fed gardiquimod, spanning more than two orders of magnitude (~20 μM to 5 mM; Fig. 4c). The theophylline, *trans*-zeatin, aciclovir, and noscapine biosensors exhibit more modest activation ratios up to 2.1 for Theo-421 (Fig. 4d), 1.6 for TZea-927 (Fig. 4e), 2.0 for Acic-145 (Fig. 4f), and 1.8 for Nosc-441 (Fig. 4g). Higher activation ratios are associated with gardiquimod biosensors, likely due to lower basal expression and higher ligand affinity than that of the theophylline, aciclovir, noscapine, and *trans*-zeatin biosensors. Our results demonstrate that DRIVER-selected biosensors can be directly applied to regulate gene expression as RNA switches with no further engineering of sensor sequences to function in vivo (Fig. 4a).

A mutational analysis workflow was used to rapidly identify variants with improved in vivo activation ratios. The majority of biosensors generated in the first three DRIVER selections had cleavage fractions in the −ligand and +ligand conditions roughly symmetric ~50% (Supplementary Figure 6), which was expected since a selection that alternates between positive and negative rounds equally weights fraction cleaved (−ligand) and fraction uncleaved (+ligand). However, the robust in vivo activity of RNA switches is more highly dependent on high cleavage activities when the ligand is not present than the converse. The S4 selection included a modification of the ratio of positive to negative selection for the purpose of increasing fraction cleaved (−ligand) and resulted in sensors with higher activation ratios in vivo. Thus, shifting fraction cleaved (−ligand) of DRIVER-selected biosensors higher may be useful to improve their activation ratios. To find sequences with such shifts, we performed a comprehensive mutational analysis on of the validated noscapine, (S)-reticuline, and aciclovir biosensors to measure the impact on cleavage fraction of all single-base substitutions, insertions, deletions, and some two-base modifications. Each biosensor sequence was subjected to mutagenic PCR and the resulting libraries were assayed with CleaveSeq in the presence or absence of the respective ligand (Supplementary Figure 10). We found that several of the mutations increased fraction cleaved (−ligand) and improved fold change of cleavage fraction between the −ligand and +ligand conditions; for example, positions 16 and 19 of the SRet-584 biosensor, possibly due to improved interactions between the two stem loops in the −ligand condition resulting in increased cleavage. These modified biosensors were then tested in vivo and found to have improved activation ratios. For example, the DRIVER-selected (S)-reticuline biosensor, SRet-584, had a fraction cleaved (−ligand) of 82% and in vivo activation ratio of 3.4. A single-base mutation at position 16 (SRet-499) was identified by the mutation screen, resulting in increased fraction cleaved (−ligand) of 95% and an increased in vivo activation ratio of 4.2 (Supplementary Figure 10a and Supplementary Data 3). Similarly, the Nosc-311 sensor, exhibited a fraction cleaved (−ligand) of 90% and activation ratio of 1.2, and when modified via a single-base mutation at position 19 (Nosc-441) exhibited an increased fraction cleaved (−ligand) and activation ratio of 93% and 1.8%, respectively (Supplementary Figure 10c). The results highlight the power of the in vitro mutational analysis to shift the operation point of biosensors such that they exhibit greater in vivo switch performance, without requiring in vivo optimization.

Finally, DRIVER-selected (S)-reticuline biosensors were applied to detect intracellular changes in metabolite concentrations, supporting applications in enzyme discovery or evolution, or dynamic feedback control in metabolic engineering efforts[15,33]. We validated the ability of biosensor-regulated fluorescent reporters to sense intracellular metabolite production in an engineered production host (Fig. 4h). (S)-reticuline biosensors were cloned into the dual-reporter construct described above and transformed into a yeast strain that was engineered to express the heterologous BIA pathway to produce the key branchpoint intermediate (S)-reticuline[34–37]. To modulate (S)-reticuline production levels through the pathway, the sensors were characterized in the presence and absence of 5 mM L-DOPA, a precursor substrate that is metabolized via several enzymatic steps to produce (S)-reticuline (Supplementary Figure 11). (S)-reticuline sensors SRet-499, SRet-046, and SRet-674 responded to the upstream feeding of L-DOPA, exhibiting activation ratios ranging from 3.6 to 4.2 (Fig. 4i). Liquid chromatography-mass spectrometry (LC-MS) analysis indicated that extracellular (S)-reticuline accumulated to levels of $2.11 \pm 0.05$ and $7.3 \pm 0.1 \, \mu M$ (Supplementary Figure 11f) in the absence and presence of 5 mM fed L-DOPA, respectively, and that pathway intermediates were present at much lower relative levels (Supplementary Figure 11f). Note that these sensors did not exhibit any response to L-DOPA in vitro (Supplementary Figure 7e) and no significant switching was observed in response to L-DOPA in a strain without the (S)-reticuline biosynthetic pathway (Supplementary Figure 11e). Feeding L-DOPA across a range of concentrations resulted in a graded response of relative fluorescence levels, indicating that the (S)-reticuline biosensors respond in a concentration-dependent manner (Fig. 4j). These data demonstrate that DRIVER-selected biosensors can be applied to monitor metabolite production levels in biosynthetic pathways.

## Discussion

We have demonstrated a fully automated, parallelized selection method (DRIVER) that can be run against unmodified individual small molecules or complex mixtures, significantly expanding our capability to generate biosensors to diverse small molecules. We further developed CleaveSeq as a high-throughput cleavage assay to functionally characterize enriched libraries and identify hits from the selection. Both methods incorporate a key improvement in the post-RT ligation step to achieve two important enhancements over previous methods—robust library regeneration and scalability by automation. DRIVER identified RNA biosensors to small molecules that range from synthetic small-molecule drugs to more structurally complex plant secondary metabolites and hormones. The biosensors exhibit a range of ligand sensitivities, spanning high micromolar to low nanomolar binding affinities, and display high selectivity against other similar ligands. Furthermore, DRIVER is an in vitro evolution platform that effectively selects new biosensors that are directly functional in living cells, with activation ratios up to 33-fold demonstrated in regulating gene expression in yeast. Finally, (S)-reticuline biosensors were applied to monitor different levels of the accumulated product from a biosynthetic pathway in an engineered yeast strain. Taken together, our results support the potential for the DRIVER platform to scalably generate biosensors with a broad range of ligand sensitivities and selectivities as required by downstream applications.

DRIVER is capable of iteratively enriching biosensor libraries for hundreds of cycles to extensively explore library sequence space without introducing undesired amplicons. The typical approach to SELEX is to incorporate partitioning methods with high enrichment efficiencies to reduce the number of rounds and avoid amplicons. However, earlier work[22,23] on joint aptamer–biosensor discovery methods identified a basic limitation on enrichment—after the first several rounds of selection, sequences appear that are not responsive to the ligand. These sequences were shown to represent ligand-insensitive RNA that cleave under the selection conditions roughly 50% of the time, likely representing sequences that fold into different conformations, some cleaving and others not, such that during each round of selection a subset survives. Although sequences that display ligand-responsive differences in cleavage fractions have a selection advantage over ligand-insensitive sequences and will eventually overtake them, the presence of these intermediate ligand-insensitive sequences limits the enrichment rate to a maximum of 2 × /round for a "perfect" switch (i.e., one with 100% cleavage in the absence of ligand and no cleavage in the presence of ligand). For switches with cleavages similar to our observed DRIVER-selected biosensors, enrichment would be ~1.6 × /round. DRIVER addresses these low enrichment rates by fully automating the process such that enrichments of $10^{12}$ or more can be readily obtained in a 5-day run. Thus, DRIVER's combination of automation and efficient regeneration makes performing a high

number of rounds to account for low enrichment efficiencies associated with joint aptamer–biosensor discovery both practical and productive.

The DRIVER workflow depends on a few parameters that can be altered in a given selection experiment in order to tune various properties in the evolved biosensors, including sensitivity for target ligands, basal cleavage fraction, and selectivity against analogs. When applied as gene regulatory switches, biosensors with a large activation ratio in vivo require high ligand sensitivity and low basal expression as a result of fast self-cleavage. The gardiquimod and reticuline switches exhibited the largest activation ratios of the biosensors characterized in vivo, and exhibit lower basal gene expression levels and greater ligand affinity. The (S)-reticuline switches with greater sensitivities were evolved by gradually lowering ligand concentration as the selection progressed, likely resulting in higher affinities of the resulting biosensors. During transcription, the low, physiological $Mg^{2+}$ concentration maintained is likely critical to evolve biosensors that undergo efficient self-cleavage and switching for in vivo function[33]. In addition, the selection rounds that generated the gardiquimod sensors were adjusted to maintain low-$Mg^{2+}$ concentrations at the RT step, which makes the selection conditions more similar to in vivo conditions during the entire process. This property is likely important for lowering in vivo basal expression, which is important for greater dynamic range in vivo. The ratio of selection rounds for cleaving sequences to rounds for non-cleaving sequences can be increased to bias the enrichment for fast-cleaving sequences. The gardiquimod switches, which display some of the fastest-cleaving sequences and highest sensitivities, were evolved by iterative cycles consisting of two rounds of cleaver selections for every one round of non-cleaver selection. We hypothesize that the lower $Mg^{2+}$ concentration during RT and the greater number of selection rounds for cleaving sequence together contribute to the exceptional performance of gardiquimod sensors relative to other ligands. However, without multiple, systematic rounds of testing the effect of different parameters in the DRIVER experimental conditions, these are hypotheses for future investigation. Furthermore, different small-molecule ligands have different conformations and functional groups, which likely impose different limitations and requirements on the sequence and structure of a biosensor, such that the occurrence of ligand-sensitive sequences may be more or less rare in a random library. Finally, while most DRIVER-selected biosensors demonstrate high selectivity against close analogs that were not used in target mixtures, negative counterselections can be performed in future work to tailor selectivities as desired against specific molecules.

While our results demonstrate the ability of the DRIVER platform to generate novel sensors and their immediate application as gene control switches in vivo, there are limitations to the system and the sensors derived from it. For example, the DRIVER-selected theophylline sensors do not display as high of an in vivo dynamic range as earlier sensors designed with a known aptamer and optimized through an in vivo screening assay[27]. The previously reported in vivo quantitative screens are likely better suited for further improving the performance of sensors specifically for cellular function. The DRIVER-selected theophylline sensor also displays lower binding affinity, which is likely due to the high concentration (10 mM) of theophylline ligand used during the DRIVER selection. Because the theophylline sensor selection was a proof-of-concept study during the initial development of DRIVER, we did not spend substantial effort in optimizing the theophylline selection conditions, such as lowering the concentration of the ligand during selection to increase the stringency of selection to give rise to sensors with greater sensitivity. However, due to the low magnesium concentrations used during the co-transcriptional cleavage incubation step of the DRIVER process, the theophylline sensor selected herein has little or no dependence on magnesium concentration, while the binding affinity of the original theophylline aptamer used in previously published sensors shows a strong dependence on magnesium concentration[38]. Further, while DRIVER enables selection for RNA biosensors against complex ligand mixtures, we only found biosensors to a subset of the tested ligands. The success rate for generating specific, high-affinity biosensors is possibly ligand dependent, as the number of selection cycles required for different ligands ranged from as early as 36 rounds for (S)-reticuline to ~126 rounds for trans-zeatin, indicating a different preponderance of aptamer sequences in the initial library for different ligands. However, we may not have sampled a large enough sequence space to rule out potential aptamers for any ligand. The robustness and scalability of the DRIVER platform will enable extensive exploration of library sequence space and systematic examination of the various factors underlying the selection outcome in future work.

Although DRIVER-selected biosensors can be directly applied as gene-regulatory switches, as shown in this study, further optimization of the gene-regulatory activities may be desired, depending on the application. We demonstrated that mutational analysis of individual parent sensor sequences can be performed to improve the in vivo gene regulatory activity of the original sensor sequence. The aptamer regions can also be isolated from DRIVER-selected biosensors and integrated into alternative RNA gene-regulatory platforms, which may exhibit improved activities in specific cell types, including microRNAs[31] and ribosome binding sites[39], or other ribozymes from that used here, including hammerhead, twister, or hepatitis delta virus ribozymes[40]. Finally, a number of high-throughput cell-based assays have been described, including RNA-sequencing[41] and FACS-seq[27], that efficiently optimize the gene-regulatory activity of an RNA switch built on the ribozyme platform used here given an identified aptamer sequence. These massively parallel cell-based assays can be applied efficiently to DRIVER-selected biosensors to further optimize their performance in desired cell types.

RNA biosensors have been used as genetically encoded controllers for a variety of synthetic biology applications[42,43]. Expanding the diversity of ligand-responsive RNA switches can increase the number and complexity of logical operations in biological circuits[44–46]. In biomanufacturing, biosensors that can monitor the accumulation of intermediate or product metabolites can be applied to generate screens of selections for improved enzyme activity or pathway flux[15,47] or to implement dynamic feedback control[46]. Biosensors that respond to small-molecule drugs with safe pharmacological profiles (e.g., aciclovir, gardiquimod) can enable the generation of conditional genetic controllers for CRISPR/Cas9-based therapies[29], gene therapies[48], and cell therapies[49,50] to mitigate toxic effects of prolonged transgene expression or cell activation[12]. In addition, the application of multiple DRIVER-selected biosensors sensitive to different ligands or to different ligand concentrations coupled with CleaveSeq can be used as a quantitative, multiplexed assay that can report on ligand concentrations in solution. In short, there is no lack of applications for aptamer-based biosensors[42,43], but there has been a severe bottleneck in generating new sensor domains relevant to the envisioned application space. The combination of DRIVER automation, ligand multiplexing, and the high-throughput CleaveSeq assay enable a strategy that provides high scalability for generating biosensors de novo that can substantially expand the repertoire of biosensors available. Our ability to detect a greater set of chemical diversity will advance a wide range of applications requiring in vivo or in vitro biosensing.

## Methods

**Library design and template preparation.** The selection process, as shown in Fig. 3a, begins with the design and synthesis of degenerate libraries that contain adequate diversity to explore the sequence space of aptamers and associated RNA switches. Libraries were constructed based on the sequence of the sTRSV hammerhead ribozyme by replacing the wild-type loop I and II sequences with a randomized sequence. One loop was replaced with either 30 or 60 random nucleotides, while the other loop was replaced with between four and eight nucleotides. This resulted in 20 different library designs, each with the wild-type sTRSV catalytic core and stem helices, and 34–68 degenerate positions (Supplementary Figure 2). The ribozyme sequence was then bracketed by A-rich prefix and suffix spacer sequences[2] (prefix spacers "W," suffix spacer "X"; Supplementary Table 2) designed to minimally interact with other sequences in the ribozyme and prepended with the T7 promoter sequence. Although the diversity of the sequence space is quite high (~$10^{41}$) and the initial library can only access a small subset of this space (~$10^{14}$), each potential aptamer can exist in many contexts within the library (e.g., there are ~$10^{26}$ possible contexts for any given 25-nucleotide sequence with this space).

The libraries were synthesized as reverse-complement DNA oligonucleotides of the desired RNA sense sequence and PAGE-purified by Integrated DNA Technologies with hand-mixed degenerate positions adjusted for coupling efficiency to achieve equal representation of the four bases (Supplementary Table 2; BT1165–1175p, BT1321–1330p). Four equimolar mixtures of the oligonucleotides were formed, each consisting of the five oligonucleotides with the same long-loop configuration. These were then annealed with the forward T7 promoter sequence (Supplementary Table 2; BT88p) by mixing at 10 μM each, heating to 95 °C in a duplex buffer (Integrated DNA Technologies) and then cooling at 0.5 °C/s to produce four template libraries for the run-off T7 transcriptions performed during the first round of selection.

**Solution-based RNA biosensor selection.** Solution-based biosensor selection was performed as a series of rounds coupling cleavage and amplification reactions. Each selection round preferentially amplified either sequences that resulted in self-cleavage of the associated RNA or sequences that did not cleave (Fig. 3a). Rounds that amplified RNA sequences that did not cleave were performed in the presence of the target molecules.

Each selection round began with a T7 transcription reaction of the library template or the prior round product under the following conditions: 20–100 nM template, 1× RNApol buffer, 9 mM ribonucleoside tri-phosphates (rNTPs), 5 U/μl T7 RNA polymerase (New England Biolabs), 1 U/μl SUPERase In (Thermo Fisher Scientific), and 10 mM dithiothreitol (DTT). The excess rNTPs over standard T7 polymerase conditions result in chelation of most of the free $Mg^{2+}$, providing a rough approximation to sub-millimolar cellular $Mg^{2+}$ concentrations, thereby making the selection conditions more representative of in vivo cellular conditions and reducing the rate of ribozyme cleavage. A reference oligonucleotide (Supplementary Table 2; BT1180p) was added to the transcription reaction at 5 nM to provide an absolute concentration reference for downstream quantitative PCR (qPCR) reactions. The transcription reactions were incubated at 37 °C for 15–30 min, during which time the transcribed RNA may undergo self-cleavage depending on the catalytic activity of the particular library sequence.

The RNA products from the transcription reaction were immediately transformed to cDNA in a RT reaction. The RNA products were diluted 2× and mixed with a reverse primer at 2 μM. Annealing of the RT primer to the RNA partially unfolds the ribozyme, thereby stopping the cleavage reaction. For uncleaved selection rounds, the RT primer consisted of the reverse complement of the expected RNA sequence from the 3′ leg of the stem II helix through the "X" spacer (Supplementary Table 2; BT575p). For cleaved selection rounds, the RT primer was prepended with an additional sequence to assist in the subsequent ligation step (Supplementary Table 2; BT1316p for rounds that started with a "Z" prefix, BT1508p for those with a "W" prefix). This mixture was diluted 2× into an Omniscript (Qiagen) RT reaction following the manufacturer's instructions and incubated at 50 °C for 20 min followed by heat inactivation at 95 °C for 2 min. The reaction products were then slow-cooled to 25 °C at 0.5 °C/s to allow refolding of the cDNA.

The products from the RT reaction include cDNA copies of both cleaved and uncleaved RNA, the latter with the prefix intact. For uncleaved selection rounds, the RT reaction mixture was diluted 40× into a HotStart Taq (New England Biolabs) PCR reaction (1 mM $MgCl_2$, 0.2 mM dNTPs, 2 ng/μl Salmon Sperm DNA, 400 nM primers) using primers that bind to the prefix (Supplementary Table 2; "W" prefix, BT1285p, for rounds that started with a "W" prefix; "Z" prefix, BT1510p for those with a "Z" prefix) and "X" suffix (Supplementary Table 2; BT575p). These prefix primers include the T7 promoter region such that the resulting PCR product can be used in a subsequent selection round. The PCR reactions were run for 5–9 cycles (under the following conditions: 95 °C for 30 s, 57 °C for 30 s, AND 68 °C for 30 s) and provided an estimated amplification of ~8×. The use of Taq to amplify the library at this stage introduced some random mutagenesis into the library, further expanding accessible diversity.

For rounds in which the cleaved reaction products were amplified, a ligation reaction was performed to ligate the 3′ end of the cDNA formed from the cleaved product to a new prefix sequence. Each of the RT primers used in this step has a 5′

end with an additional sequence and is 5′-phosphorylated (Supplementary Table 2; BT1316p with a "W" prefix, BT1508p with a "Z" prefix), such that it acts as both a splint and a substrate for the subsequent ligation step as shown in Fig. 2b. Since the cDNA from the uncleaved RNA already has a prefix, this cDNA will not align correctly with the self-splinting RT primer and thus not be ligated during the reaction. The RT reaction mixture was diluted 2.5× into a T4 ligation reaction with the addition of 1× T4 Ligase Buffer and 2 U/μl T4 Ligase Enzyme (New England Biolabs) and incubated 15 min at 37 °C followed by heat inactivation at 65 °C for 10 min.

Following the ligation reaction, the cDNA from the cleaved RNA will be circularized as shown in Fig. 1b. The self-splint primers were synthesized with Uracil bases at select locations (Supplementary Table 2; BT1316p, BT1508p), such that a Uracil-Specific Excision Reagent (USER, New England Biolabs) will cut the cDNA at these locations, thereby releasing the desired product and permitting amplification of this product via PCR. The PCR conditions used to amplify the USER-cleaved cDNA products were as described above for the uncleaved cDNA product amplification, except that USER was added to the reaction components at a concentration of 0.01 U/μl and the reaction was incubated at 37 °C for 10 min prior to the first cycle of the PCR.

Selection series (S1, S2, S3, and S4) began with libraries that used the "W" prefix. Several rounds (Supplementary Table 1) of cleaved selection as described above were performed in the prefix alternating between "W" and "Z" for each round. Following these cleavage-selection rounds, the targets were introduced and selection proceeded with a mix of uncleaved selection rounds with target present and cleavage-selection rounds without target present.

**Automation of selection steps and assays.** Selection steps, CleaveSeq assays, and NGS library preparations were automated on a Tecan EVO-150 liquid handling system configured with a thermocycler (T-Robot, Biometra), reagent chiller (RIC20XT, Torrey Pines), robotic arm (RoMa, Tecan), and shaker (BioShake 3000elm, QInstruments). Reactions were performed using 96-well microplates (Eppendorf DNA LoBind), sealed during PCR and incubations with Microseal P + Pads (Bio-Rad). Pipetting was performed using PTFE Teflon-coated stainless-steel tips. After each use, tips were washed with 2% sodium hypochlorite and then rinsed with Milli-Q water. Custom Python software was used to drive and monitor the robot control. The automation platform and code enabled 8–12 selection rounds per day of unattended selection with multiple independent selections with different target mixtures running concurrently.

**An NGS assay for simultaneous measurement of fraction cleaved of individual sequences in a ribozyme library (CleaveSeq).** The CleaveSeq protocol followed the same method as the cleaved selection rounds described above, up to the PCR step. At this point, the reaction was split and run through two separate PCR reactions, one that amplified the cleaved components with a "Z" prefix and the other that amplified the uncleaved components with a "W" prefix. The primers used in the above PCR reactions included 5′-overhang regions with Illumina adapters and barcodes to allow each read to be identified as to the assay conditions. In addition to the standard Illumina index barcodes embedded in the adapters, we also added 1–10 nucleotides of custom barcode nucleotides between the Illumina adapters and the prefixes or suffixes (Supplementary Table 2; "NGS Primer"). The variable length barcodes introduce shifts of otherwise identical sequence positions in the prefix and suffix regions of the DNA being sequenced, resulting in more equal distribution of the four nucleotides at each position. This strategy improves the performance of Illumina sequencers' clustering step, which relies on distinct sequences in adjacent clusters during the first 20 sequencing cycles. Also, at the input to the barcoding step, 15 distinct reference sequences, each with either a "W," "Z," or "A" prefix, with five different lengths, and all similar to the library structure, are spiked-in at a fixed 18 pM concentration each (Supplementary Table 2; "NGS reference"). During the analysis, the number of reads of reference sequences provides a conversion factor for equating the number of reads with absolute concentration. The PCR reaction mixtures (1× Kapa HiFi enzyme, 1× Kapa HiFi buffer, 400 nM primers) were run for 18 cycles (under the following conditions: 98 °C for 30 s, 57 °C for 30 s, and 72 °C for 30 s).

The barcoded libraries were mixed in ratios based on the relative number of reads desired for each library and the libraries were diluted to 4 nM of DNA with Illumina adapters as quantified by qPCR (KAPA Library Quantification Kit). PhiX was spiked into the sequencing library at 10–20% of the total library concentration to further improve the cluster calling of the Illumina pipeline for amplicons. The libraries were sequenced on an Illumina platform, either MiSeq (using MiSeq Control software v3.0), NextSeq (using NextSeq Control software v2.1.0), or HiSeq using 2 × 75 or 2 × 150 reads, depending on the data needs of a particular experiment, in each case using Illumina recommended loading guidelines.

**Analysis of NGS data to determine cleavage fractions.** Analysis of NGS data was performed by a custom pipeline. The steps consisted of paired-end alignment using PEAR[51], grouping identical reads to form a table of unique sequences and number of reads observed for each one, assignment of a unique accession number to each ribozyme sequence (with the prefix/suffixes removed), and insertion of these data into a MySQL (version 5.6) relational database (RDBMS). The

assignment of unique accession numbers allows comparison of results across multiple experiments and sequencing runs. The RDBMS allows flexible queries of the number of reads of a given ribozyme sequence or reference for each prefix and barcoded condition. The references spiked-in at a known concentration prior to barcoding allow computation of absolute concentrations of each sequence with each prefix.

The cleavage fraction of a sequence is computed as the ratio of the concentration of the sequence with the prefix corresponding to cleaved molecules ("W", "Z", or "A" depending on the steps used to create the library) to the total concentration of that sequence. That is, the cleavage fraction, $c_s$, for a particular sequence, $s$, is computed using:

$$c_s = \frac{r_{Z,s}/r_{Z,\text{ref}}}{r_{W,s}/r_{W,\text{ref}} + r_{Z,s}/r_{Z,\text{ref}}} \quad (1)$$

where $r_{P,s}$ are the number of reads of sequence $s$ with prefix $P$ and $r_{P,\text{ref}}$ are the number of reads of the reference sequences with prefix $P$. The fold change of the cleavage fraction, $f_s$, under different ligand conditions (e.g., +target or −target) is calculated as:

$$f_{s,\text{target}} = k \frac{1 - c_{s,+\text{target}}}{1 - c_{s,-\text{target}}} \quad (2)$$

Unlike the simple ratio of the cleavage fractions, this formulation gives a fold change that should be predictive of the gene-regulatory activity ratio of the biosensor when used in vivo, where only uncleaved RNA molecules result in gene expression of the intact RNA. The factor $k$ is used to compensate for slight variations in the experimental conditions in the two assays (i.e., −target and +target) that are unrelated to target presence. This factor is set to a value such that the median fold change over all sequences measured in the same run is 1.0 (since only a small fraction of sequences within each library are sensitive to any particular ligand).

Standard errors and confidence intervals for cleavage fraction, $c_s$, and fold change of the cleavage fraction, $f_s$, are dependent on the number of reads of each prefix in each condition and are computed using 1000 bootstrap samples drawn with replacement from the observed reads. Each bootstrap sample is used to compute estimates $\hat{c}_s$ and $\hat{f}_s$. The 5% and 95% percentiles of the $\hat{c}_s$ and $\hat{f}_s$ are then used as the confidence intervals for $c_s$ and $f_s$.

Complete software implementations of the above analysis pipeline are available (see Code availability statement).

**Biosensor hit identification assays.** The CleaveSeq assay can be used to identify sequences that are sensitive to a target within a library following selection. However, at all but the later rounds of a selection where particular sequences have significantly enriched, the libraries contain diversity much higher than the number of sequences that can be read using NGS, resulting in only very few reads of any given sequence. In order to have sufficient reads per sequence to estimate the fraction cleaved of that sequence, the library is first constricted to a random, low-diversity subset of sequences. Constriction was performed by diluting a library to ~8 fM (~5000 molecules/µl) in salmon sperm DNA (1 ng/µl, New England Biolabs) in low-binding tubes (LoBind, Eppendorf), mixing thoroughly (~30 s vortex), and then adding 1–10 µl of that dilution to a PCR mixture (1× Kapa HiFi enzyme, 1× Kapa HiFi buffer, 400 nM primers BT1285p, BT575p; Supplementary Table 2). These steps were performed in a dedicated pre-PCR area to avoid contamination with other templates. The PCR mixture was amplified for 18 cycles (under the following conditions: 98 °C for 30 s, 57 °C for 30 s, and 72 °C for 30 s). The resulting constricted library has an approximately uniform representation of 5000–50,000 sequences randomly chosen from the full library. Note that this constricted library is used solely for analysis of the composition of a particular round of selection and is not used as input to subsequent selection rounds as that would artificially decrease diversity.

The constricted library (or, after sequencing had shown significant enrichment had occurred, the unconstricted library) was used as input to CleaveSeq assays in the presence and absence of the targets of interest. For each condition, two independent replicates were assayed starting with the RNA transcription (in the presence or absence of target) through barcoding and sequencing. NGS analysis as described above was performed on each replicate and the resulting $c_s$ were compared to identify deviations greater than expected sampling noise (Fig. 2c). The replicates were then pooled by ligand condition and the fold changes of cleavage fraction of each sequence in the constricted library were computed. Sequences with fold changes of cleavage fraction that were both >2.0 and significant at the $p = 1/N$ level, where $N$ is the number of sequences measured, were flagged as potential sensors. Sets of switch sequences for each ligand were hierarchically clustered by sequence similarity to identify unique families of sensors that have distinct consensus sequences (Supplementary Figure 4). A subset of these were selected for further validation, where preference was given to sequences that encode putative aptamers with low sequence similarity to previously validated sequences (Supplementary Data 3).

**Validation of biosensor candidates using CleaveSeq.** DNA templates for the individual putative biosensors were synthesized as two overlapping oligonucleotides (Integrated Device Technologies), with the "W" prefix and "X" suffix sequences added onto the ends. The oligonucleotides were annealed at 50 µM each

in a duplex buffer (Integrated DNA Technologies) for 2 min at 95 °C followed by slow cooling at 0.5 °C/s. The hybridized oligonucleotides were diluted to 500 nM and a three-cycle PCR (1× Kapa HiFi enzyme, 1× Kapa HiFi buffer, 0.3 mM dNTPs) was run (under the following conditions: 30 s at 98 °C, 30 s at 57 °C, and 25 s at 72 °C) to extend the 3′ ends of the oligonucleotides to form double-stranded DNA. The PCR products were then diluted to 1 nM and a second PCR (1× Kapa HiFi enzyme, 1× Kapa HiFi buffer, 0.3 mM dNTPs) was run for 11 cycles (under the following conditions: 30 s at 98 °C, 30 s at 57 °C, and 25 s at 72 °C) using PAGE-purified primers BT1285p and BT575p (Supplementary Table 2). The products from this second PCR reaction were cleaned up using Ampure XP beads at 1.8× ratio of bead reagent volume to PCR reaction volume according to the manufacturer's instructions.

All identified biosensors and controls (Supplementary Data 3; sTRSV, sTRSVCtl, Grz_xxx) were equimolar mixed and then the mixture was assayed at ~2 nM of the total template using CleaveSeq in each target condition with three replicates per condition and analyzed using the custom NGS analysis pipeline as described above. Cleavage fractions, fold changes of cleavage fractions, and their standard errors were computed from the mean and standard deviation of the estimates over the replicates.

**Gel electrophoresis characterization of ribozyme cleavage fractions.** Denaturing PAGE was performed on biosensor RNA to assess cleavage activity. Double-stranded DNA templates prepared as previously described were used in transcription reactions: 20–100 nM template, 1× RNApol buffer, 9 mM rNTPs, 5 U/µl T7 RNA polymerase (New England Biolabs), 1 U/µl SUPERase In (Thermo Fisher Scientific), and 10 mM DTT. Transcript reactions were run for 30 min at 37 °C. The transcription and cleavage reactions were stopped by the addition of EDTA to a final concentration of 4 mM. A measure of 2× TBE-urea sample loading buffer (Bio-Rad) was added to each reaction mixture. Samples were denatured by incubating at 95 °C for 10 min and loaded onto a denaturing 10% polyacrylamide gel with 8 M Urea in a Mini-PROTEAN® Electrophoresis System (Bio-Rad). Electrophoresis was performed at 240 V for 45 min. The gel was subsequently stained with the GelRed loading dye (Biotium) for 10 min before imaging with the ethidium bromide setting directly on a GeneSys G:Box fluorescent gel imager from Synoptics (Frederick, MD, USA). The FIJI implementation of ImageJ, version 2.0.0-rc-68/1.52e, was used to quantify the intensities $I$ of the gel bands. The fraction cleaved, $f$, was determined by:

$$f = \frac{\frac{(I_{\text{cleaved}} - I_{\text{bg}})}{\text{length}_{\text{cleaved}}}}{\frac{(I_{\text{cleaved}} - I_{\text{bg}})}{\text{length}_{\text{cleaved}}} + \frac{(I_{\text{uncleaved}} - I_{\text{bg}})}{\text{length}_{\text{uncleaved}}}} \quad (3)$$

and the fold change of the fraction cleaved was determined by:

$$\text{fold change} = \frac{1 - f_{+\text{ligand}}}{1 - f_{-\text{ligand}}} \quad (4)$$

where $I_{\text{bg}}$ is background intensity.

**Measurement of the effect of point mutations on cleavage activity of biosensors.** Selected biosensors were further analyzed to determine the effect of point mutations on cleavage fraction and fold change of cleavage fraction. Double-stranded DNA templates prepared as previously described were subjected to 15 cycles of mutagenic PCR with Mutazyme II Enzyme (Genemorph) with primers BT1285p and BT575p (Supplementary Table 2) following the manufacturer's protocol. The resulting mutagenesis library was run through the CleaveSeq assay and custom Matlab software (see Code availability statement) was used to compute the change in cleavage fraction and fold change of cleavage fraction for each single-nucleotide mutation observed. For purposes of presentation, secondary structure prediction was performed using NUPACK[52] to identify the lowest energy conformation exhibiting the same stem-loop configuration as the native sTRSV ribozyme (Supplementary Figure 10).

**SPR measurement of the binding affinities of RNA biosensors.** To prepare DNA templates for transcription (see Supplementary Table 2 for oligonucleotide sequences), all PCR reactions in this section used the Kapa HiFi HotStart PCR Kit (Roche) and 400 nM each of two primers, performed for ten cycles with 10 nM starting template concentration, at an annealing temperature of 55 °C, using the GC buffer and 1 M betaine monohydrate, unless otherwise specified. To prepare DNA templates of biosensors for SPR binding assays, the previously prepared double-stranded templates as described in the CleaveSeq validation assays were amplified using primers BT1285p and JX457 to append the T7 promoter and the poly(A) sequence for hybridizing transcribed RNA molecules to the poly(T) sequence on the sensor chip. The JX457 primer also incorporates a G12A mutation into the catalytic core of the ribozyme to prevent the transcribed RNA from cleaving during the SPR assay. Non-binding negative control sequences used were the starting libraries with randomized loop I and loop II sequences, with the architecture that each biosensor was derived from, and prepared with PCR amplification using BT1285p and JX457 as above. To replace the small stem loop II with N5 for TZea-579, Nosc-441, Acic-145, and Theo-421, BT480, which contains a degenerate N5 in stem loop II, was used along with BT1285p as primers to PCR

amplify the template. To replace the small stem loop I with N5 for TZea-927 and SRet-760, the template was amplified using BT469, which contains a degenerate N5 in stem loop I, and JX457. The full-length sequences were also synthesized as one piece to make TZea-927.LoopIN5 and SRet-760.LoopIN5 for comparison, and the eventual results were identical. These loop-randomized sequences were diluted 1:25 into another PCR reaction with BT1285p and JX457 for eight cycles to generate the DNA template for transcription. To truncate the binding domain of each of the biosensors to the large stem loop, a one-piece full-length oligonucleotide (in reverse complement) was synthesized to flank the stem-loop sequence with a T7 promoter at the 5′ and poly(A) sequence at the 3′, of the eventual DNA template (Supplementary Table 2). A double-stranded DNA template was generated with PCR amplification of the synthesized oligonucleotide with mJX1 and mJX2. All PCR products were purified using the DNA Clean & Concentrator Kit (Zymo Research) according to the manufacturer's instructions, and the concentration of the PCR product was quantified on a NanoDrop (Thermo Fisher Scientific). Transcription was performed using the MEGAshortscript™ T7 Transcription Kit (Thermo Fisher Scientific) according to the manufacturer's instructions, using 100 nM DNA template and incubating at 37 °C for 5 h. Transcribed RNA was purified using the RNA Clean & Concentrator Kit (Zymo Research) according to the manufacturer's instructions and quantified on a NanoDrop.

Binding affinity characterizations of transcribed RNA biosensors were performed using an SPR assay on the Biacore X100 instrument from GE Healthcare (Chicago, IL, USA) as previously described[8]. Briefly, a Biacore CM5 sensor chip (GE Healthcare) was immobilized with a poly(T) sequence 5′-/5AmMC6/TTTTTTTTTTTTTTTTT TTTTTTTTTTTTTTTTTTTT-3′ (Integrated DNA Technologies), using the Amine Coupling Kit (GE Healthcare) according to the manufacturer's instructions. The running buffer used was HBS-N (GE Healthcare) for the SPR assay, which was made up of 0.1 M HEPES, 1.5 M NaCl, pH 7.4, and was supplemented with 0.5 mM $MgCl_2$ (Thermo Fisher Scientific). The transcribed RNA was diluted into the running buffer to provide ~2.5 μg RNA per cycle in the run protocol. Typically, the multicycle kinetics protocol was performed where RNA was regenerated for every cycle when ligand was associated onto and dissociated from the sensor chip surface once. The multicycle kinetics protocol consisted of (1) capturing the RNA onto the sensor chip with 40 s contact time at a 5 μl/min flow rate, (2) associating and dissociating the ligand for 180 s each at a 30 μl/min flow rate, and (3) regenerating the sensor chip using 25 mM NaOH with a 30 μl/min flow rate for 30 s. For sequences that demonstrated very slow off rates, as was the case for the (S)-reticuline biosensors, a single-cycle kinetics protocol was performed with identical conditions as above, except that the same RNA was used for the consecutive association of the ligand at five concentrations before a long (>30 min) ligand dissociation. All SPR measurements are taken for three or more replicates except for the gardiquimod biosensor, due to shelter-in-place lockdown measures preventing further experimentation to provide replicate data. Biacore X100 Control software (version 2.0.2) was used to acquire the data.

All analyses were performed using a custom MATLAB script available at https://github.com/jsxiang/analyzeSPR. Analyses were performed on raw data exported using Biacore X100 Evaluation software (version 2.0.2). To reduce nonspecific ligand effect in determining the binding affinity constants, the SPR sensorgram curves (in response units, RUs) in flow cell 2 were first subtracted by the signal in the reference flow cell 1 (no RNA captured) to give the FC2-1 SPR curves. Next, the FC2-1 SPR curves for cycles with ligands were background subtracted by the blank cycle, where running buffer without ligand was flown over the captured RNA. Finally, the blank subtracted SPR curves of the negative control were used to subtract that from the sample biosensors to give the normalized SPR RU, $RU_{norm}$. A Langmuir model[53] was fit to $RU_{norm}$ to determine the ligand on and off rates, $k_{on}$ and $k_{off}$. The fit was sometimes impossible to determine (designated non-measurable), due to on or off rates being too fast (<2 s for change from baseline to saturated SPR RU or vice versa) for the sensitivity of the instrument. The kinetic dissociation constant was determined by taking the ratio, $K_D$ (kinetic) $= \frac{k_{off}}{k_{on}}$. Equilibrium binding affinity constants, $K_D$ (equilibrium), were determined by taking the mean of a 5-s window after the SPR curves $RU_{norm}$ have reached saturation, and evaluating the inflection point of a sigmoidal curve fit. All estimated parameters ($k_{on}$, $k_{off}$, $K_D$(kinetic), $K_D$(equilibrium)) were reported as the mean ± s.e.m. across three or more replicates (Supplementary Data 4). Sequences that did not demonstrate sensorgram-like binding curves with ligand-concentration-dependent increases in RU, but resembled the negative control were designated non-binding.

**Validation of RNA biosensors as gene-regulatory switches in yeast**. The gene-regulatory activities of RNA biosensors (except those responsive to (S)-reticuline) in yeast cells were assayed as previously described[27]. Biosensors were PCR amplified with Gibson overlapping primers sJX105 and sJX18 (Supplementary Table 2) from double-stranded DNA templates constructed as described in the CleaveSeq assay section. The PCR products were cloned via Gibson assembly into the 3′-UTR of the *GFP* reporter gene in pCS1748 (Supplementary Figure 11g), which was digested with *Avr*II and *Xho*I restriction enzymes (New England Biolabs). Sequence verified and purified plasmids were transformed into yeast strain W303α using the Frozen-EZ Yeast Transformation II Kit (Zymo Research) according to the manufacturer's instructions and plated on YNB–URA dropout plates. After 2 days of incubation at 30 °C, colonies were inoculated in liquid YNB–URA dropout media for 16 h overnight growth with

shaking at 250 r.p.m. at 30 °C before back-diluting 40-fold to an approximate $OD_{600}$ of 0.05 into fresh YNB–URA and growing for 6 h to $OD_{600}$ ~0.4, in the presence and absence of ligand. The yeast cells were diluted 20-fold into 1 × PBS supplemented with 1% bovine serum albumin before assaying on a MACSQuant VYB flow cytometer from Miltenyi (Bergisch Gladbach, Germany), with forward scatter of gain 275 V, side scatter of gain 262 V, and using lasers with excitation wavelengths 561, 405, and 488 nm in channels Y2 (filter 615/20 nm; gain 399 V), V1 (filter 450/50 nm; gain 329 V), and B1 (filter 525/50 nm; gain 468 V), respectively (Supplementary Figure 11a–c). MACSQuantify software version 2.8 was used to acquire the data.

Biosensors responsive to (S)-reticuline were similarly cloned and transformed into yeast cells as described above. The yeast strain used was CSY1171, a CEN.PK2-derived strain with several native gene knockouts and heterologously expressed genes in the BIA pathway to overproduce (S)-reticuline[36]. Transformed colonies were grown for 16 h overnight with shaking at 250 r.p.m. at 30 °C in YNB–URA dropout media and cultures were back-diluted 100-fold into fresh YNB–URA dropout media containing 10 mM ascorbic acid and 0 or 5 mM L-DOPA, which were both freshly dissolved and sterile filtered. Cultures were grown at 30 °C with shaking at 250 r.p.m. for 24 h before assaying for fluorescence levels via flow cytometry with the same settings as above.

During analysis, cells were gated for viable and singlets, and transformed cells (mCherry > $10^{3.2}$ and GFP > $10^{3.1}$ fluorescence units) were used in downstream analysis. For each sequence assayed in pCS1748, the fluorescence intensity of $i$ cells in the GFP channel was normalized by the fluorescence intensity in the mCherry channel to give the mean fluorescence ratio of a sample,

$$F_{sample} = mean\left(\frac{GFP_i}{mCherry_i}\right) \quad (5)$$

To determine the relative fluorescence unit, the mean fluorescence ratio of the sample, taken across three or more transformation replicates, was normalized to that of the non-cleaving sTRSV ribozyme mutant sTRSVctl, and scaled by a factor of 100 as follows:

$$RFU = \frac{F_{sample}}{F_{sTRSVctl}} \times 100 \quad (6)$$

This normalization was performed to remove nonspecific fluorescence effects due to the addition of ligands, and to standardize measurements across different experiments that may be subject to fluctuations in instrument gain (Supplementary Data 2). All flow cytometry results were analyzed using a custom MATLAB script, available at https://github.com/jsxiang/FlowAnalysis.

YNB–URA media samples were taken for analysis by liquid chromatography with tandem mass spectrometry (LC-MS/MS) with an Agilent 1260 Infinity Binary HPLC and an Agilent 6420 Triple Quadrupole LC-MS to determine the accumulation of (S)-reticuline and upstream 1-BIA intermediates, as previously described[35]. LC-MS/MS: Agilent MassHunter Workstation LC/MS Data Acquisition for 6400 Series Triple Quadrupole software (ver. B.08.02) was used to acquire the data. Samples were separated on an Agilent EclipsePlus C18, 2.1 50 mm, 1.8 μm column with 0.1% formic acid as solvent A and acetonitrile with 0.1% formic acid as solvent B at a constant flow rate of 0.4 ml/min and an injection volume of 5 μl. For separation of the compounds, the following method was used: 0–0.1 min, 7% B; 0.1–5 min, 10–35% B; 5–5.5 min, 35–90% B; 5.5–7 min, 90% B; 7–7.01 min, 7% B; followed by a 3 min equilibration at 7% solvent B. The LC eluent was directed to the MS for 1–5 min with electrospray ionization source gas temperature 350 °C, gas flow of 11 L/min, and nebulizer pressure 40 PSI. For quantification, the MS was used in multiple reaction monitoring (MRM) mode. The MRM transitions used for quantification are 272 → 107 for norcoclaurine, 286 → 107 for coclaurine, 300 → 107 for N-methylcoclaurine, 316 → 192 for 3′-hydoxy-N-methylcoclaurine, and 330 → 137 for (S)-reticuline, all using fragmentor 135 and collision energy 25[35]. Quantification of (S)-reticuline was based on integrated peak area MRM chromatogram using the Agilent MassHunter Workstation and reported as the mean ± s.e.m. of three biological replicates based on generated standard curves for (S)-reticuline. LC-MS/MS: Agilent MassHunter Workstation Qualitative Analysis Navigator software (ver. B.08.00) was used to perform the integrated peak area analysis.

**Reporting summary**. Further information on research design is available in the Nature Research Reporting Summary linked to this article.

## Data availability

Biosensor sequences are available through GenBank (accession numbers appear in Supplementary Data 3). Aptamer and other short sequences are provided in the Supplementary information. NGS data are available at DOI 10.6084/m9.figshare.13697490. A step-by-step CleaveSeq method is also available[54]. All other data are available from the authors on request. Source data are provided with this paper.

## Code availability

Custom software used for analyses is available at github.com/btownshend and github.com/jsxiang.

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

## Acknowledgements

We thank Matias Kaplan, James Payne, and Ben Kotopka for valuable feedback in the preparation of this manuscript and Maureen McKeague for her insights into aptamers and SPR. This work was supported by funds from the National Institutes of Health [grant to C.D.S.], Agency for Science, Technology, and Research [fellowship to J.S.X.], and E.I. Dupont De Nemours and Company.

## Author contributions

B.T. and J.S.X. have contributed equally to this work. B.T. and C.D.S. conceived the project. B.T., J.S.X., and C.D.S. wrote the manuscript. B.T., J.S.X., G.M., and E.H. conducted the experiments. The software for robot control, NGS, and qPCR analysis was developed by B.T. The software for flow cytometry analysis and SPR analysis was developed by J.S.X., B.T., J.S.X., and C.D.S. designed the experiments and analyzed the results.

## Competing interests

The authors declare the following competing interests: patent application US16/884,941.
