## [Peer Review File · Nature Communications]

Reviewers' Comments:

Reviewer #1:

Remarks to the Author:

In this manuscript the authors present a new method for nucleic acid aptamer biosensor evolution termed DRIVER. The authors present the novel concept of selection which involves selecting for cleaved and non-cleaved sequences in the presence of a small molecule. The DRIVER approach enriches RNA sequences which self-cleave in the presence of the target. The approach can be automated and scaled. The authors demonstrate against six small molecule targets then show application for biosensing as in vivo biosensors for gene-regulatory switches in yeast.

The underlying idea for DRIVER selection system is highly original. There have been some related approaches using splint methods prior, but this is the first time a non-splint approach has been taken to allow bias-free selection. The authors then go on to show how effective the approach is rigorously with 6 targets, with well-presented datasets. When coupled to the application in yeast, I would argue that the work meets the threshold for a significant advance with multidisciplinary impact suitable for Nature Communications.

I do have some comments for improvement in the manuscript:

1. I would recommend a second biophysical approach (eg microscale thermophoresis) to determine affinities for the small molecules. In particular, this is necessary for noscapine for which the binding data is problematic (Fig S8C) and for theophylline where both on and off rates are extremely fast and SPR is not an ideal approach.
2. Surface plasmon resonance data. The authors appear to be only using the equilibrium position to determine their K_D 's but it would be informative to properly analyse the on and off rates. The kinetics appear to vary extensively from extremely fast (theophylline) to data which should fit well (reticuline) to entirely problematic (noscapine). The on and off rates are important to consider for in vivo application yet appear to be entirely ignored. For weak binding in particular one needs control experiments with closely related molecules (those used in Fig S7) Figure S8: For Biacore show full on and off rates (table in Fig S9b seems to fit better with Fig S8 than within Fig S9) and check how they correlate with the equilibrium K_D 's (which appear to be the ones shown?)
3. Figure S7a It would be more simple to correlate the selectivity with reference to the structures of acyclovir/ganciclovir in the figure (also all other sub-panels). Challenging to correlate next to text on page 9. Likely S7 would get too busy, so recommend split into greater number of supplementary figures also showing molecular structures.
4. Fig 4h is hazy.
5. Fig 4k – this is not mentioned in main text anywhere, unclear what this is showing.

Julian A. Tanner

Reviewer #2:

Remarks to the Author:

This manuscript presents a directed evolution pipeline to generate and characterize RNA biosensors derived from the hammerhead ribozyme. This pipeline consists of two components – DRIVER, an automated selection method to enrich for sequences that permit or forbid cleavage based on the presence or absence of ligand, and CleaveSeq, an NGS-based strategy to determine cleavage efficiency by read count in a population. Sensors derived from this pipeline were then tested for their ability to regulate gene expression in vivo via a two color assay in yeast. The strategy presented is well put-together and logically sound. While this method does improve upon

existing biosensor development strategies in terms of throughput and scalability, it was interesting to see that the generated biosensors did not have very high dynamic ranges. Nevertheless, there are many methodological advances that would be interesting and useful for a range of researchers in this area. We recommend this work for publication with minor revisions after addressing the comments discussed below.

Major Comments

1) What is the cleavage threshold for determining whether or not a sequence is a biosensor, ie, is there some concrete number that can be used to determine if a selection has yielded a sensor or not? Extending this, what cleavage threshold distinguishes a "good" from a "bad" sensor?

2) Figures 4D-E: The DRIVER-derived switches for theophylline, trans-zeatin, Aciclovir, and nescapine all show in vivo dynamic ranges of ≤ 2 , significantly lower than those reported for Gardiquimod. The heatmap in figure 3E similarly suggests that the Gardiquimod switches performed exceptionally well relative to the other ligands. Why is this? This discrepancy in performance suggests that DRIVER's ability to generate sensors with a high dynamic range is contingent on some property of the target ligand, making it less generalizable than claimed.

3) Following on point 3: It would be useful if the authors expanded the discussion about how the choice of ribozyme platform, or aptamer fusion point (i.e. library design) might have affected the end results of biosensor performance. Would the authors anticipate that a different type of RNA mechanism (for example the conditional CRISPR guide RNAs or the other mechanisms mentioned) would generate more high performance biosensors? What would be needed to adapt the methodology to select for these mechanisms instead? This information would be very useful for users wishing to use this approach more broadly.

Minor Comments

4) Can the authors compare the resulting DRIVER-selected theophylline switches to previous theophylline aptazymes in terms of performance? This would be interesting.

5) It would be good if the heat map data in Figure 3e could be displayed using a bar chart in the SI or as an excel data file for quantitative comparisons if needed.

Response to Reviewer Comments:

We would like to thank the reviewers for their constructive comments and helpful suggestions for improving our manuscript. The comments were very useful in helping us to strengthen the manuscript and clarify the presentation of our work. In the sections below, we address each comment made by the reviewers.

Response to Reviewer #1 Comments:

In this manuscript the authors present a new method for nucleic acid aptamer biosensor evolution termed DRIVER. The authors present the novel concept of selection which involves selecting for cleaved and non-cleaved sequences in the presence of a small molecule. The DRIVER approach enriches RNA sequences which self-cleave in the presence of the target. The approach can be automated and scaled. The authors demonstrate against six small molecule targets then show application for biosensing as in vivo biosensors for gene-regulatory switches in yeast. The underlying idea for DRIVER selection system is highly original. There have been some related approaches using splint methods prior, but this is the first time a non-splint approach has been taken to allow bias-free selection. The authors then go on to show how effective the approach is rigorously with 6 targets, with well-presented datasets. When coupled to the application in yeast, I would argue that the work meets the threshold for a significant advance with multidisciplinary impact suitable for Nature Communications.

We thank the reviewer for the encouraging remarks and address the individual comments below.

I do have some comments for improvement in the manuscript:

1. I would recommend a second biophysical approach (eg microscale thermophoresis) to determine affinities for the small molecules. In particular, this is necessary for noscapine for which the binding data is problematic (Fig S8C) and for theophylline where both on and off rates are extremely fast and SPR is not an ideal approach.

We thank the reviewer for the suggestion to use microscale thermophoresis as an independent biophysical approach. Although it has taken some time for us to locate and obtain access to an instrument during the ongoing pandemic, we were able to gain limited access to a Nanotemper Monolith device and ran numerous sets of measurements in an attempt to characterize their equilibrium K_{DS} . Despite assistance from Nanotemper's application engineers and multiple troubleshooting attempts, the data do not show any measurable thermophoresis effect for any of the tested aptamer/small molecule combinations. In addition, we attempted to use MST to characterize the well-known TCT8-4 theophylline aptamer and again saw no measurable thermophoresis (see attached slides for representative MST results). From these results and consultation with a colleague that has also employed MST for small molecule-RNA aptamer analysis with similar results, we believe that the aptamers for which we have attempted MST do not undergo a conformational or other change that results in a change in thermophoresis greater than the noise floor of the instrument. It is possible that an untested condition (e.g., buffer or additive) may result in a usable MST signal, but running exhaustive testing in another department's laboratory and equipment during

these times has been difficult due to limitations in place at the university. In addition, microscale thermophoresis does not produce on/off rates, providing only equilibrium K_D 's, and thus would not improve the on/off time estimates.

We do however note that our results already include three independent measurements that confirm binding of the aptamers with the ligands: (i) the biophysical measurements using SPR, (ii) the indirect measurements obtained using CleaveSeq, and (iii) the activity of the sensor *in vivo*. As shown in Figure 3d, these independent measurements are consistent with each other.

As noted by the reviewer, the fast on/off rates for some of the aptamers, such as for theophylline and noscapine, are poorly characterized by SPR. Although it would theoretically be possible to measure higher on-rates with SPR by decreasing the concentration of the ligand, the low masses of the ligands produce only a small response (<20 RU), and lowering the concentrations result in the noise floor overwhelming the signal. To address this, we did attempt to use a higher density SPR chip (i.e., CM7 instead of CM5) to increase the signal to access lower ligand concentrations, but this did not result in a higher density of captured RNA. In any case, even for these aptamers, SPR is quite capable of determining equilibrium K_D 's. We have included a new table in Supplementary Figure S8, which shows that the equilibrium K_D and kinetic K_D measurements are highly similar for the sensors for which the kinetics were measurable.

We acknowledge the reviewer's concern regarding the noscapine SPR data, which showed some deviation from expected curves. In particular, as can be seen in Figure S8c, the noscapine sensorgrams systematically show a decrease in signal during the association phase and we assume that the reviewer's comment about noscapine being problematic arises from this decrease. Under the assumption that these decreases were due to a noscapine-induced release of the captured RNA from our polyT DNA linker, we made several changes to our protocol and ran additional SPR experiments. These changes included increasing the length of the polyT linker, using a sequence complementary to the X suffix of our sensors, and replacing some of the nucleotides of the linker with locked nucleic acid analogues; all of which should increase the stability of the DNA:RNA hybrid. These changes did not result in sensorgrams without a decrease in signal during the association phase, and the cause of this decrease is still unknown.

If, after consideration of the above points, the editor believes an additional biophysical validation is required for publication, we can attempt to find resources to run isothermal titration calorimetry (ITC) or a kinetic exclusion assay (KinExA). Each of these has their own limitations and whether they can be successfully applied to measure these binding constants is by no means clear. Furthermore, they are unlikely to provide more than an additional validation of the equilibrium K_D s. Indeed, the difficulty in measuring binding of unmodified small molecules to RNA aptamers led to our choice of SPR as the most effective method available.

*2. Surface plasmon resonance data. The authors appear to be only using the equilibrium position to determine their KD 's but it would be informative to properly analyse the on and off rates. The kinetics appear to vary extensively from extremely fast (theophylline) to data which should fit well (reticuline) to entirely problematic (noscapine). The on and off rates are important to consider for in vivo application yet appear to be entirely ignored. **For weak binding** in particular one needs control experiments with closely related molecules (those used in Fig S7) Figure S8: For Biacore show full on and off rates (table in Fig S9b seems to fit better with Fig S8 than within Fig S9) and check how they correlate with the equilibrium Kd 's (which appear to be the ones shown?)*

We thank the reviewer for these comments. We have revised the main text to discuss the binding kinetics of the biosensors where the kinetic measurements were possible. Specifically, (*S*)-reticuline biosensors display the slowest off rates, and this may contribute to the high activation ratio observed *in vivo*.

In answer to the reviewer's concern about the binding affinity measurements for biosensors with weak binding, we agree that control experiments are important. For negative controls, we used non-binding sequences, which were the starting libraries with randomized loop I and loop II sequences in the architecture that each biosensor was derived from. In the data analysis, we subtract the SPR response of each tested biosensor by the SPR response of the negative control for the same ligand at the same concentration to obtain a normalized SPR response for downstream binding kinetics and affinity analyses, as described in the methods section. We believe the randomized loop sequences are closely related negative controls. Experiments with closely related molecules would be assessing specificity, and we believe the specificity measurements made using CleaveSeq achieve that goal.

We have included a new table (Figure S8g) in the revised manuscript that displays binding kinetics for the sensorgrams in Figure S8 and thank the reviewer for this suggestion. We also include a brief discussion of the correlation of the kinetic K_{DS} with the equilibrium K_{DS} in the main text.

3. Figure S7a It would be more simple to correlate the selectivity with reference to the structures of acyclovir/ganciclovir in the figure (also all other sub-panels). Challenging to correlate next to text on page 9. Likely S7 would get too busy, so recommend split into greater number of supplementary figures also showing molecular structures.

We thank the reviewer for this suggestion. We have revised Figure S7 to show the molecular structures alongside each selectivity chart as suggested. The revised presentation of this figure has clarified the data.

4. Fig 4h is hazy.

We thank the reviewer for pointing this out. Figure 4h (and Figure 4a) have been replaced with higher resolution versions in the revised manuscript.

5. Fig 4k – this is not mentioned in main text anywhere, unclear what this is showing.

We thank the reviewer for catching this typo. The reference in the text was incorrect and should have referenced Figure 4k, but had indicated 4j. The manuscript text has been revised accordingly.

Response to Reviewer #2 Comments:

*This manuscript presents a directed evolution pipeline to generate and characterize RNA biosensors derived from the hammerhead ribozyme. This pipeline consists of two components – DRIVER, an automated selection method to enrich for sequences that permit or forbid cleavage based on the presence or absence of ligand, and CleaveSeq, an NGS-based strategy to determine cleavage efficiency by read count in a population. Sensors derived from this pipeline were then tested for their ability to regulate gene expression *in vivo* via a two color assay in yeast. The strategy presented is well put-together and logically sound. While this method does improve upon existing biosensor development strategies in terms of throughput and scalability, it was interesting to see that the*

generated biosensors did not have very high dynamic ranges. Nevertheless, there are many methodological advances that would be interesting and useful for a range of researchers in this area. We recommend this work for publication with minor revisions after addressing the comments discussed below.

We thank the reviewer for the encouraging remarks and address the individual comments below.

Major Comments

1) What is the cleavage threshold for determining whether or not a sequence is a biosensor, ie, is there some concrete number that can be used to determine if a selection has yielded a sensor or not? Extending this, what cleavage threshold distinguishes a “good” from a “bad” sensor?

We thank the reviewer for this question, which brings up several interesting points. The classification of a given sensor as “good” or not ultimately depends upon the application and the requirements for that application.

For the general work of identifying potential sensors independent of their ultimate application, we determine that a sequence is a sensor if the null hypothesis of the cleavage of a particular sensor being insensitive to the target molecule is rejected, as described in the caption for Figure 3b. We then characterize the sensor in terms of fold change of cleavage, typically treating sensors with a fold change of cleavage greater than two as “good” ones.

For use *in vivo*, the criterion translates to one of activation ratio. Theoretically, a sensor with 50% *in vivo* basal expression corresponds to a 2-fold sensor if the presence of ligand leads to full inhibition of self-cleavage. Based on the data shown in Figure 4k, this corresponds to a cleavage fraction of around 50%. The exact threshold for a ‘good’ sensor would likely be in the range of 2-fold to 20-fold, depending on the desired activation ratios needed in the particular application, with lower cleavage fractions resulting in a greater range of potential activation ratios.

2) Figures 4D-E: The DRIVER-derived switches for theophylline, trans-zeatin, Aciclovir, and noscapine all show in vivo dynamic ranges of ≤ 2 , significantly lower than those reported for Gardiquimod. The heatmap in figure 3E similarly suggests that the Gardiquimod switches performed exceptionally well relative to the other ligands. Why is this? This discrepancy in performance suggests that DRIVER’s ability to generate sensors with a high dynamic range is contingent on some property of the target ligand, making it less generalizable than claimed.

We thank the reviewer for this comment. We revised the text in the discussion section to clarify why we believe the gardiquimod sensors perform well relative to the sensors generated to other ligands. Specifically, the selection rounds that generated the gardiquimod sensors were adjusted to maintain low Mg²⁺ concentrations at the reverse transcription step, which makes the selection criteria more stringent for the fraction cleaved and gave rise to sensors with greater fraction cleaved. This property is likely important for lowering *in vivo* basal expression, which is important for greater dynamic range *in vivo*. However, without multiple, systematic rounds of testing the effect of different parameters in the DRIVER experimental conditions, these are hypotheses for future investigation. We recognize that different small molecule ligands have different conformations and functional groups which likely impose different limitations and requirements on the sequence and structure of a biosensor, such that the occurrence of

such sequences may be more or less rare in a random library. Thus, we have included a related note about ligand-specific properties that may limit DRIVER's ability to generate sensors with high dynamic range as we have not rigorously ruled out this possibility.

3) Following on point 3: It would be useful if the authors expanded the discussion about how the choice of ribozyme platform, or aptamer fusion point (i.e. library design) might have affected the end results of biosensor performance. Would the authors anticipate that a different type of RNA mechanism (for example the conditional CRISPR guide RNAs or the other mechanisms mentioned) would generate more high performance biosensors? What would be needed to adapt the methodology to select for these mechanisms instead? This information would be very useful for users wishing to use this approach more broadly.

We thank the reviewer for the suggestion to expand the discussion on the broader use of this evolution platform and have added new text to the discussion section to clarify the following. We have chosen the sTRSV hammerhead ribozyme as it is one of the most well studied ribozymes and historically we have more experience integrating it with aptamers. As new hammerhead ribozymes have recently been discovered¹, if they demonstrate faster constitutive cleavage rates, these may be readily adapted to our evolution platform via a similar library architecture, to potentially lead to biosensors with lower basal expression levels, which is important to achieving high activation ratios *in vivo*. New classes of ribozymes have also recently been discovered, though more work needs to be done to first determine optimal aptamer fusion points, as pointed out by the reviewer. Once a clear set of aptamer integration rules are delineated, an appropriate loop-around splint ligation monomer can be designed and the same principles of ligand-dependent modification to the cleavage and structure of the ribozyme can be applied in the DRIVER set up.

Regarding whether utilizing a different RNA mechanism would lead to better sensors, we believe nontrivial effort would be needed for optimization and troubleshooting DRIVER to a different mechanism and it is unclear whether a different mechanism in and of itself will inherently generate better biosensors. The process for adapting DRIVER to a different RNA mechanism depends on the type of functionality that is required to be evolved. In the example of conditional CRISPR guide RNAs, if the mechanism is aptazyme-based², where the addition of a ligand facilitates the self-cleavage of the ribozyme and processing of the guide RNA, this ligand dependent self-cleavage activity can be adapted to the DRIVER platform to evolve for novel sensing capabilities. To evolve sensors that undergo self-cleavage in the presence of ligand instead of the opposite – absence – as described in our work, the ligand can be added at the co-transcriptional cleavage step and cleaving sequences be selected, followed by a subsequent round where the non-cleaving sequences are selected in the absence of ligand. If it is desired that the context of the gRNA such as the tracrRNA portion is preserved to evolve sensors that are compatible with the gRNA, one would extend the sequence and design the corresponding primers to include this extension for the amplification steps. The self-cleavage and selective ligation aspects are core to the functionality of DRIVER, to enrich the genetic information that reflects the desired ligand sensing event, so systems that have a ligand dependent self-cleavage characteristic are generally compatible with DRIVER. It is not straightforward to adapt DRIVER to evolve other aspects of conditional CRISPR gRNA activity that depend on other molecules not physically linked to the sensing domain, e.g., a target RNA that gets cleaved, the Cas9 protein itself, which can be selectively amplified by PCR.

Minor Comments

4) *Can the authors compare the resulting DRIVER-selected theophylline switches to previous theophylline aptazymes in terms of performance? This would be interesting.*

We thank the review for this interesting question. We now include in our discussion section a comment about comparing the DRIVER-selected theophylline sensors with the ones we previously engineered and published³, which have been characterized in identical *in vivo* assays. Namely, the DRIVER-selected theophylline sensors do not display as high of an *in vivo* dynamic range as the earlier sensors optimized through an *in vivo* screening assay. The previously reported *in vivo* quantitative screens are likely better suited for further improving the performance of sensors specifically for cellular function. The DRIVER-selected theophylline sensor also displays lower binding affinity, which is likely due to the high concentration (10 mM) of theophylline ligand used during the DRIVER selection. Because the theophylline sensor selection was a proof-of-concept study during the initial development of DRIVER, we did not spend substantial effort in optimizing the theophylline selection conditions, such as lowering the concentration of the ligand during selection to increase the stringency of selection to give rise to sensors with greater sensitivity. However, due to the low magnesium concentrations used during the co-transcriptional cleavage incubation step of the DRIVER process, the theophylline sensor selected herein has little or no dependence on magnesium concentration, while binding affinity of the original theophylline aptamer used in previously published sensors shows a strong dependence on magnesium concentration⁴.

5) *It would be good if the heat map data in Figure 3e could be displayed using a bar chart in the SI or as an excel data file for quantitative comparisons if needed.*

We thank the reviewer for this suggestion. The source data for Figure 3e (and the other figures) have been added to the revised submission as an excel data file as requested.

References:

1. Weinberg, Z. *et al.* New classes of self-cleaving ribozymes revealed by comparative genomics analysis. *Nature Chemical Biology* **11**, 606–610 (2015).
2. Tang, W., Hu, J. H. & Liu, D. R. Aptazyme-embedded guide RNAs enable ligand-responsive genome editing and transcriptional activation. *Nature communications* **8**, 15939 (2017).
3. Townshend, B., Kennedy, A. B., Xiang, J. S. & Smolke, C. D. High-throughput cellular RNA device engineering. *Nature methods* **12**, 989–94 (2015).
4. Jucker, F. M., Phillips, R. M., McCallum, S. A. & Pardi, A. Role of a heterogeneous free state in the formation of a specific RNA-theophylline complex. *Biochemistry* **42**, 2560–7 (2003).

Microscale Thermophoresis

Acic-145 Stem

MST Traces

Binding

- Acic-145 stem loop containing aptamer was chemically synthesized by IDT with a Cy5 label
- Three replicates bracketing the expected KD of the aptamer (700nM) were run using 40nM RNA in Tris buffer with 0.5mM MgCl₂
- Range of mean response < 3 normalized fluorescence units without any clear ligand concentration dependence

Microscale Thermophoresis

TCT8-4 Theophylline Aptamer (Med)

MST Traces

Binding

- TCT8-4 was transcribed in vitro and annealed to 30nM polyT DNA sequence chemically synthesized by IDT with a Cy5 label
- 3 replicates bracketing the expected KD of the aptamer (200 nM) were run using 40nM RNA
- Jenison buffer A (100mM HEPES, 500mM NaCl, 5mM MgCl₂)
- No-theophylline condition plotted as 1nM
- No clear ligand concentration dependence

Microscale Thermophoresis

TCT8-4 Theophylline Aptamer (High)

- TCT8-4 was transcribed in vitro and annealed to 30nM polyT DNA sequence chemically synthesized by IDT with a Cy5 label
- 3 replicates bracketing the expected KD of the aptamer (200 nM) were run using 40nM RNA
- Jenison buffer A (100mM HEPES, 500mM NaCl, 5mM MgCl₂)
- No-theophylline condition plotted as 1nM
- No clear ligand concentration dependence

Reviewers' Comments:

Reviewer #1:

Remarks to the Author:

The authors have made a comprehensive rebuttal to my comments in the first review.

Whilst it is unfortunate that the MST data did not work out, I am confident in the data with as the authors argue the three independent measurements by SPR, Cleaveseq and in vivo activity. The additional Supplementary Figure S8 is also an improvement. Controls are better described with the new table S8g.

Figures S7 is clearer and 4k is corrected.

Overall this is an excellent paper suitable for Nature Communications which I now recommend for publication.

Julian A Tanner

Reviewer #2:

Remarks to the Author:

We have read through the response to reviewers comments and the revised manuscript. The authors have done a thorough job addressing all of our comments and we believe this manuscript will be of significant interest when published.